# Single nuclei RNA-seq of mouse placental labyrinth development

Bryan Marsh[1,2], Robert Blelloch[1,2]*

[1]The Eli and Edythe Broad Center of Regeneration Medicine and Stem Cell Research, Center for Reproductive Sciences, University of California, San Francisco, San Francisco, United States; [2]Department of Urology, University of California, San Francisco, San Francisco, United States

**Abstract** The placenta is the interface between mother and fetus in all eutherian species. However, our understanding of this essential organ remains incomplete. A substantial challenge has been the syncytial cells of the placenta, which have made dissociation and independent evaluation of the different cell types of this organ difficult. Here, we address questions concerning the ontogeny, specification, and function of the cell types of a representative hemochorial placenta by performing single nuclei RNA sequencing (snRNA-seq) at multiple stages of mouse embryonic development focusing on the exchange interface, the labyrinth. Timepoints extended from progenitor-driven expansion through terminal differentiation. Analysis by snRNA-seq identified transcript profiles and inferred functions, cell trajectories, signaling interactions, and transcriptional drivers of all but the most highly polyploid cell types of the placenta. These data profile placental development at an unprecedented resolution, provide insights into differentiation and function across time, and provide a resource for future study.

## Introduction

The placenta links maternal tissues and the embryo, providing necessary support and instructing the development of the embryo. The placenta performs many essential functions including transport of nutrients and exchange of gases and waste between maternal and fetal blood, production and uptake of hormones, and regulation of the maternal immune system (*Maltepe and Fisher, 2015*; *Woods et al., 2018*). The mouse placenta is separated into three main regions – the decidua, the junctional zone (JZ), and the labyrinth. Select fetal trophoblast invade into the maternal decidua and remodel and line maternal arteries, facilitating maternal blood to flow into the fetal portion of the placenta (*Adamson et al., 2002*; *Maltepe and Fisher, 2015*). The JZ lies directly beneath the decidua and consists of parietal trophoblast giant cells, spongiotrophoblast, and glycogen cells that largely function in hormone secretion and metabolism (*Simmons et al., 2007*; *Woods et al., 2018*). Beneath the JZ, closest to the embryo, is the labyrinth, which is specially designed to maximize surface area for gas and nutrient exchange between maternal and fetal blood (*Figure 1A*; *Simmons et al., 2008a*; *Soncin et al., 2015*; *Woods et al., 2018*).

The development of the labyrinth structure begins with fusion of the allantois with the chorion at E8.5 (*Cross et al., 2003*). This begins a process of invasion and branching morphogenesis resulting in an expansive surface area for gas and nutrient exchange by E10.5 (*Soncin et al., 2015*). The gas exchange interface in the mouse contains three layers of differentiated trophoblast separating maternal blood from the endothelial cells of the fetal vasculature (*Simmons et al., 2008a*; *Maltepe and Fisher, 2015*; *Coan et al., 2005*). Sinusoidal trophoblast giant cells (S-TGC) reside in the maternal blood space, have demonstrated exocrine functions, and are a source of placental lactogens (*Simmons et al., 2008b*). S-TGC are attached to the outermost syncytiotrophoblast layer (SynTI) but are perforated allowing maternal blood to contact SynTI (*Coan et al., 2005*;

*For correspondence:
robert.blelloch@ucsf.edu

Competing interests: The authors declare that no competing interests exist.

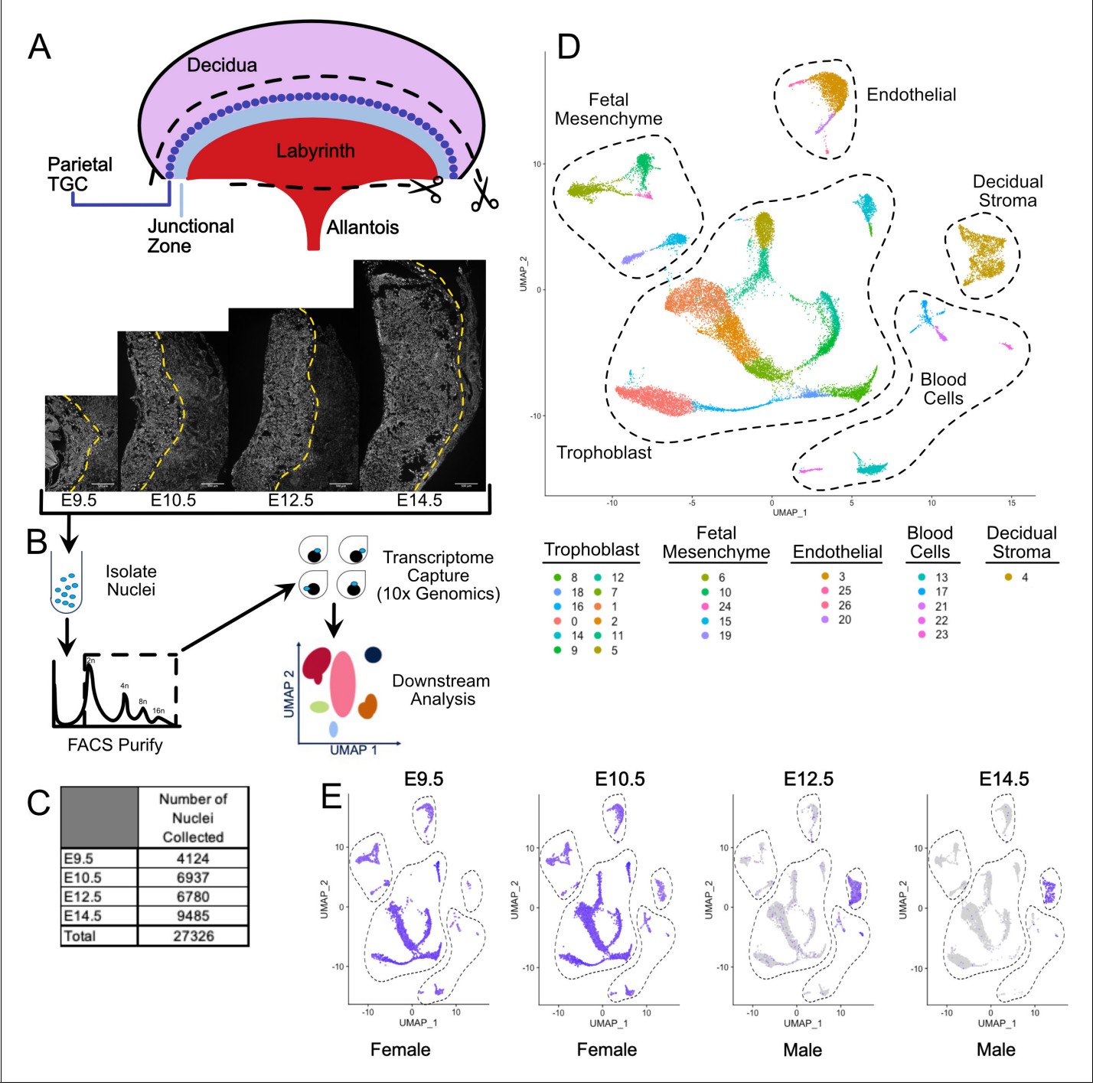

**Figure 1.** Nuclear isolation and snRNA-seq of mouse placental cells (E9.5-E14.5). (**A**) Schematic showing the main regions of the placenta – Labyrinth, Junctional Zone, Parietal TGC, and Decidua. Removal of the decidual stroma and the allantois is marked by scissors and cut lines. Sections of placentas at E9.5–14.5 stained with DAPI to label nuclei and reveal tissue architecture. The dotted line is drawn at the interface between the junctional zone (left of the line) and the maternal decidual stroma (right) to demonstrate the growth occurring during this time span. (**B**) Schematic outlining nuclei isolation from tissue, purification using FACS, transcriptome capture, and downstream analysis. (**C**) Number of nuclei collected at each gestational age and in total. (**D**) Visualization of the 27,326 nuclei included in the analysis plotted in two dimensions by transcriptome similarity using uniform manifold approximation and projection (UMAP). Each dot represents one nucleus colored according to assignment by clustering analysis (see Materials and methods). Dotted lines encircle clusters with common properties. (**E**) Expression of *Xist* in each sample. Female placenta samples express *Xist* in all nuclei. Male placental samples express *Xist* only in maternally derived nuclei. The main cell type groupings from D are outlined here also. The online version of this article includes the following source data and figure supplement(s) for figure 1:

*Figure 1 continued on next page*

*Figure 1 continued*

**Source data 1.** Marker genes for clusters in complete snRNA-seq data (E9.5-E14.5).
**Figure supplement 1.** Quality control metrics andmarker genes for snRNA-se of mouse placentae (E9.5-E14.5).

*Woods et al., 2018*). Next, are the two SynT layers which together transport gasses and nutrients from the maternal blood to the fetal vasculature (*Georgiades et al., 2002*; *Woods et al., 2018*). SynTI is the outer layer in direct contact with maternal blood forming the barrier between maternal blood and fetal tissue. Beneath, is SynTII which functions as an intermediary between SynTI on its apical membrane and the fetal endothelium on its basal side. Proper patterning of each cell type ensures efficient transport of nutrients and waste between maternal and fetal blood, and is necessary for embryonic growth, development, and viability.

Investigating the cells of the placental labyrinth is difficult due to their size and an expansion in the number or ploidy of nuclei. SynT cells are multinucleate and are not amenable to standard dissociation and isolation procedures. As a reference, human SynT are 80–200 µM in diameter and contain 5–15 nuclei meaning their size and fragility do not allow for profiling by FACS or passage through microfluidic channels for droplet-based single-cell RNA sequencing (*Liu et al., 2018*). Therefore, questions remain about how each cell type is specified, how essential placental functions are distributed between cell types, and how each cell type changes throughout developmental time.

Single-cell RNA sequencing has become the preferred method for understanding the composition of a complex tissue at the transcriptional level. Several studies in both mouse (*Nelson et al., 2016*; *Home and Ghosh, 2019*) and human (*Vento-Tormo et al., 2018*; *Suryawanshi et al., 2018*) have profiled the placenta using scRNA-seq, however, syncytiotrophoblast are greatly underrepresented in these data. Studies in mouse have either avoided profiling the labyrinth trophoblast, or are unable to distinguish between the two SynT layers (*Nelson et al., 2016*; *Home and Ghosh, 2019*). Recently, application of droplet-based transcriptome profiling to nuclei has been shown to provide high quality data and results which recapitulate those from whole cell scRNA-seq (*Habib et al., 2017*; *Lake et al., 2018*; *Bakken et al., 2018*; *Ding et al., 2020*). To circumvent complications with single-cell dissociation and capture of placental cells, we applied single nuclei RNA-seq (snRNA-seq) to the mouse placenta.

In this study, we used snRNA-seq to profile the murine placental development at four time points spanning placental cell type specification, differentiation, and maturation (E9.5, E10.5, E12.5, and E14.5). We resolve all populations of the labyrinth at single-nucleus resolution and identify distinct markers and transcript signatures for each population. We are able to separate each SynT layer, furthering the understanding of the independent roles of each cell type at the gas exchange interface. Mapping differentiation from stem/progenitor cells to terminal differentiation, we capture novel intermediates and identify putative regulators. We predict signaling interactions between populations of the labyrinth to understand crosstalk between these populations, revealing factors guiding concerted development of SynT and fetal endothelium. Finally, we identify transcription factor (TF) driven gene regulatory networks active in each placental population, uncovering candidate regulators of fate and function in the placenta. By separating and characterizing placental cell types, specifically those in the labyrinth, these data provide a resource for understanding development in vivo, generation of molecular tools in vitro and in vivo, and for future modeling of prenatal pathologies.

## Results

### Nuclear isolation and snRNA-seq of mouse placental cells (E9.5-E14.5)

To gain a better understanding of the constitution of cell types and developmental trajectories in the mouse placenta, we performed single nuclei droplet-based sequencing of their transcriptomes at four embryonic time points: E9.5, E10.5, E12.5, and E14.5. Given the largely syncytial nature of the placenta, we reasoned that sequencing of nuclei would bypass the problems inherent in using a single-cell RNA-sequencing approach. To decrease contribution from maternal tissues, the placenta was carefully dissected at the boundary between the fetal junctional zone and the maternal decidua (*Figure 1A*). On the labyrinth side of the placenta, the placenta was separated from embryo proper at the point of attachment with the allantois on the basal side (embryo facing). Nuclei were isolated

and sorted for DNA content by FACS enabling the removal of doublets and contaminating cytoplasmic RNA (*Figure 1B*). Isolated nuclei were subjected to droplet-based sequencing using the 10x Genomics platform.

A total of 27,326 nuclei passed quality control (between 500 and 4000 unique genes identified and fewer than 0.25% mitochondrial reads), ranging from 4124 to 9485 nuclei across the embryonic days (*Figure 1C*, *Supplementary file 1*). Clustering of nuclei using the Seurat package identified 26 clusters, then projected in UMAP expression space allowing for an understanding of the relationship among populations as visualized by proximity (*Figure 1D*, *Figure 1—figure supplement 1A–C*; *Satija et al., 2015*; *Butler et al., 2018*). Based on general markers, these populations were separated into five broad groups: trophoblast, decidual stroma, blood cells, endothelial cells, and fetal mesenchyme.

The maternal decidual stroma was identified by the absence of Xist expression in male embryonic tissue (*Figure 1E*). Marker genes further subdivided the five broad groups into subgroups (*Figure 1—figure supplement 1D*, *Figure 1—source data 1*, *Supplementary file 2*). The fetal mesenchyme group consisted of two distinct subgroups. One (clusters 6, 10, and 24) expressed markers of pericytes (e.g. *Acta2*) and several growth factors including *Wnt5b* and *Pdgfrb* that are critical for development and maintenance of vascular populations (*Eaton et al., 2020*). The other (clusters 15 and 19) expressed *Gata4*, *Kit,* and *Pdpn.* The nature of the cells in this subgroup is unclear, but immunofluorescence for PDPN showed the cells to be interstitial with long projections that appear to make contact with PECAM positive fetal endothelial cells (*Figure 1—figure supplement 1E*). The endothelial group consisted of cells expressing markers of vascular endothelium (Cluster 3 - *Pecam1*, *Kdr*, *Tek*), lymphatic endothelium (cluster 26 - *Flt4/Vegfr3*), and pro-angiogenic and lymphogenic lineages (Clusters 20 and 25 - *Vegfc* and *Igf1*, respectively) (*Cao et al., 1998*; *Björndahl et al., 2005*). The identification of cells expressing lymphatic markers is intriguing because the placenta is not thought to contain lymphatics. However, lymphatic endothelium has been discovered in human and it is possible a similar cell type exists in mouse (*Pique-Regi et al., 2019*). The blood cell group consisted of putative erythrocytes (Cluster 13 - Hbb-y, Hba-x), macrophages (Cluster 17 - Mrc1+; *Martinez and Gordon, 2014*), B-cells (Cluster 21 - Bank1, Btla; *Aiba et al., 2006*), T-cells (Cluster 22 – Gzmc, Slamf1, Cd84; *Griewank et al., 2007*, *Veillette, 2006*), and natural killer cells (Cluster 23 Cd*244*; *Lee et al., 2004*). Together, these data show that snRNA-seq has the sensitivity and specificity to identify the multitude of cell types in the developing placenta.

## Sub-clustering identifies the trophoblast subpopulations of the labyrinth and junctional zone

Trophoblast cells are unique to the placenta and perform the majority of its specialized functions. Unsurprisingly, they also make up the largest of the five broad groups of nuclei (*Figure 1D*). To gain a deeper understanding of the trophoblast populations, we used Seurat to cluster only trophoblast nuclei. Nuclei collected at each individual timepoint were analyzed separately and those assigned a trophoblast identity were then integrated and visualized on a single UMAP. This analysis resulted in 13 clusters consisting of 16,386 nuclei, with clusters ranging in size from 159 to 2906 nuclei (*Figure 2A*, *Figure 2—figure supplement 1A*). The clusters formed five appendages arising from a central body.

The central body consisted of 4 clusters. One of these clusters highly expressed the receptor tyrosine kinase *Met* (*Figure 2B*), which was previously described as a marker of labyrinth trophoblast progenitor (LaTP) cells with trilineage potential: SynTI, SynTII, and S-TGC (*Ueno et al., 2013*). The snRNA-seq data places these cells as most closely related to SynTII. Notably, the LaTP shared expression of the Wnt signaling pathway members *Ror2*, *Lgr5*, and *Tcf7l1* with presumptive SynTII precursors. Interestingly, the clustering identified a second subpopulation of cells in the central body that shared relatively high expression of *Tcf7l1* and the long non-coding RNA *Pvt1* with LaTPs, but expressed low levels of *Met*, *Ror2*, and *Lgr5* and high levels of *Egfr* (*Figure 2B*). These cells were more closely related to SynTI and S-TGC. We designated this subpopulation as LaTP2. Correlative analysis of RNA levels for Met and Egfr in individual cells confirmed that for the most part these markers were uniquely expressed in LaTP and LaTP2 cells, respectively (*Figure 2—figure supplement 1B*). Previously described LaTPs were also shown to express high levels of the adhesion protein EPCAM (*Ueno et al., 2013*). EPCAM mRNA and protein was mostly limited to the MET+ LaTP cells, although rare EPCAM/EGFR double positive cells were also present (*Figure 2—figure*

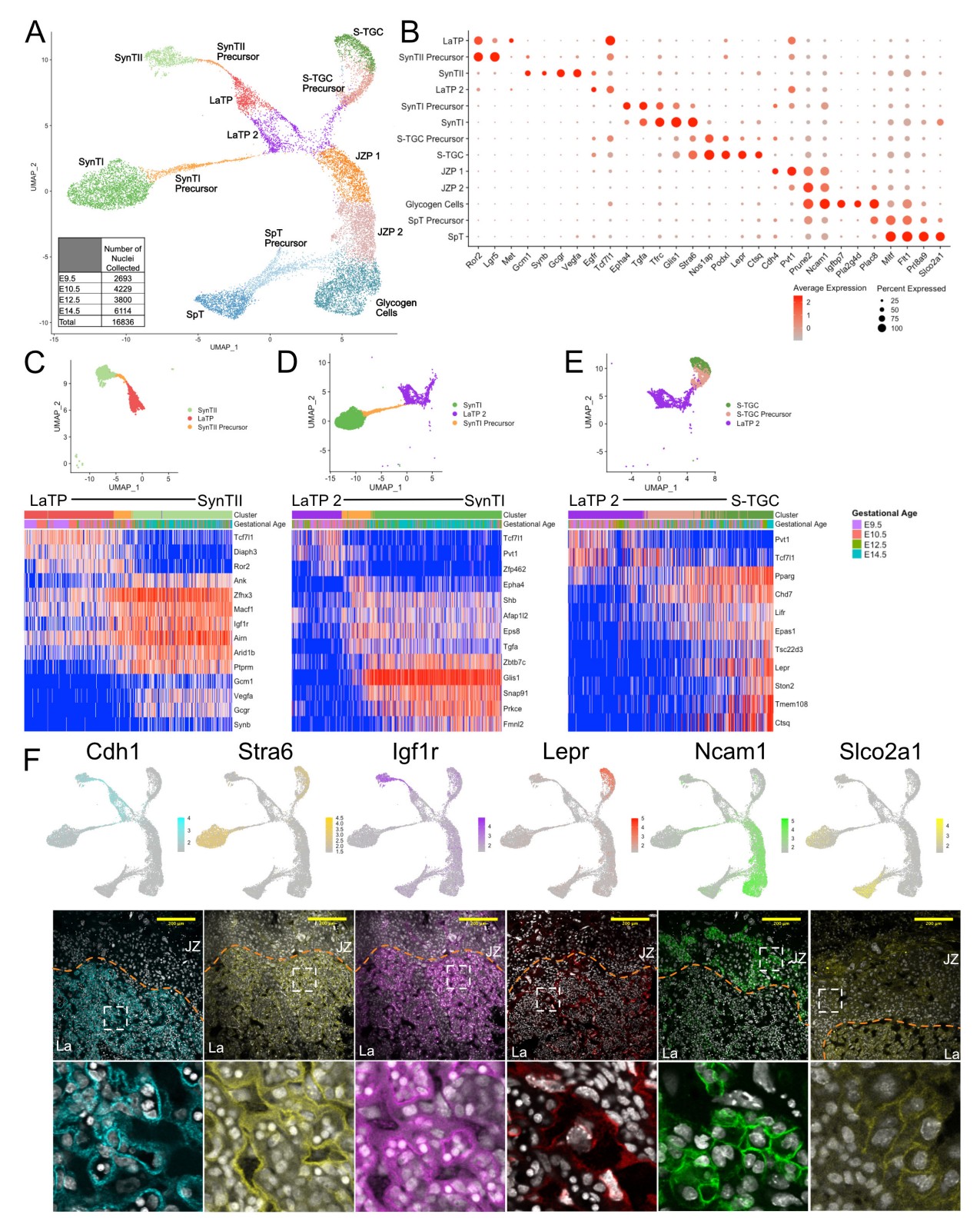

**Figure 2.** Sub-clustering of the trophoblast nuclei identifies the cell types of the labyrinth and junctional zone. (A) UMAP showing the 16,836 nuclei included in the trophoblast subset, clustered and plotted according to transcriptome similarity. Clusters were annotated according to canonical marker genes and named below. Inset shows the number of nuclei collected at each gestational age. (B) Dot plot showing average expression and percent of nuclei in each cluster expressing canonical and novel marker genes identified for each cluster. Genes listed on the x-axis and clusters on the y-axis. (C–
*Figure 2 continued on next page*

*Figure 2 continued*

E) Nuclei along the differentiation from LaTP to SynTII, SynTI, and S-TGC were ordered by pseudotime using Slingshot. The nuclei included along each pseudotime axis shown at top. The expression of select genes (y-axis) representing each differentiation are shown in each heatmap. Each column represents a nucleus organized in pseudotime proceeding from left to right along the x-axis. (F) Expression of genes unique to several trophoblast populations projected in UMAP space (top) and localization of corresponding protein in E12.5 mouse placenta sections by immunofluorescence staining (Middle – 20x, Scale Bar = 200 µM; Bottom – high magnification inset area shown by white dashed line). Separation of labyrinth (La) and junctional zone (JZ) shown by the orange dashed line.

The online version of this article includes the following source data and figure supplement(s) for figure 2:

**Source data 1.** Marker genes for clusters in subclustered trophoblast nuclei.
**Source data 2.** Expression of the top 200 variable genes along the pseudotime ordering of cells from LaTP to SynTII (*Figure 2C*).
**Source data 3.** Expression of the top 200 variable genes along the pseudotime ordering of cells from LaTP2 to SynTI (*Figure 2D*).
**Source data 4.** Expression of the top 200 variable genes along the pseudotime ordering of cells from LaTP2 to S-TGC (*Figure 2E*).
**Figure supplement 1.** Quality control metrics for snRNA-seq of subclustered trophoblast nuclei and dissection of LaTP populations by specific markers.
**Figure supplement 2.** Additional validation of the identities of Junctional Zone clusters.
**Figure supplement 3.** Characterization of spongiotrophoblast and glycogen gell clusters by specific expression of prolactin genes.

*supplement 1C–D*). Positionally, MET expressing LaTP cells were more basal than the EGFR expressing LaTP2 cells, suggesting that LaTP may be giving rise to LaTP2 (*Figure 2—figure supplement 1D*). Consistent with this interpretation, immunofluorescence staining of E8.5 embryos identified a large number of EPCAM/MET double positive cells, but few EGFR positive cells in the developing chorion plate (*Figure 2—figure supplement 1E*). Together, these data show that LaTPs can be separated into two populations based both on mRNA and protein expression. The developmental relationship between these two populations will require future lineage tracing.

The remaining two clusters within the central body appeared to represent precursors of the differentiated cells of the junctional zone – glycogen cells (GCs) and spongiotrophoblasts (SpT) [Junctional zone precursors (JZPs) 1 and 2 (*Figure 2A*)]. JZP1 exclusively expressed high levels of *Cdh4*, whereas *Prune2* and *Ncam1* showed a continuum of expression, increasing from JZP 1 to 2 and finally specified GCs (*Figure 2B* and *Figure 2—figure supplement 2A*). Markers such as *Igfbp7*, *Pla2g4d*, *Plac8, and Pcdh12* marked later stages of GC specification. In contrast to GCs, the SpT arm separated from late JZP2/early GCs, with increasing expression of *Mitf*, *Flt1*, *Prl8a9*, and *Slco2a1* as they transitioned from precursors to differentiated cells. The relationship between LaTPs and JZPs is unclear as they shared few markers. One shared marker was *Pvt1*, a target and enhancer of expression of the oncogene *Myc* and, therefore, may reflect the shared proliferative nature of these two populations, explaining their juxtaposition in UMAP space (*Tseng et al., 2014*).

The arms extending from the central body showed a transition from progenitors through precursors to fully differentiated cells representing five previously described cell types of the mouse placenta: SynTI, SynTII, S-TGCs, GCs, and SpTs. SynTI was easily identified based on the expression of known markers of SynTI function including the transferrin receptor *Tfrc* and monocarboxylase transporter *Mct1* (*Walentin et al., 2016*), but also expressed previously unknown markers such as the transcription factor (TF) *Glis1* and the retinol receptor *Stra6* (*Figure 2B*). SynTI precursors also expressed these markers, albeit at lower levels, while additionally expressing high levels of the signaling molecules *Epha4* and *Tgfa*, which decreased with further differentiation. SynTII was annotated by expression of the TF *Gcm1* and fusion gene *Synb* (*Maltepe and Fisher, 2015*; *Simmons et al., 2008a*), but also expressed many previously unknown cell-specific genes such as the glucagon receptor *Gcgr* and vascular endothelial growth factor *Vegfa*. SynTII precursors showed elevated expression of *Ror2* and *Lgr5* relative to the LaTP. In contrast, *Tcf7l1* decreased throughout the process of differentiation from LaTP to SynTII. S-TGCs were easily identified by the expression of the canonical marker *Ctsq* (*Simmons et al., 2007*; *Simmons et al., 2008a*), but also uniquely expressed the leptin receptor *Lepr* and the Nitric Oxide Synthase one adaptor *Nos1ap* among other genes. Marker genes for all populations are included in supplementary information (*Figure 2—source data 1*).

We further probed the transition from multipotent LaTP through precursors to each of the differentiated labyrinth cell types (SynTII, SynT1, and S-TGC) using pseudotime analysis. The pseudotime analysis program Slingshot aligned cells along a 2D trajectory from the most transcriptionally similar LaTP population (LaTP-SynTII; LaTP2-SynTI; LaTP2-STGC) to each of the mature populations

(*Street et al., 2018*; *Figure 2C–E*, *Figure 2—source datas 2–4*). Pseudotime analysis expanded upon the marker genes for each population by defining the timing of expression of genes through the differentiation process and identifying distinct signaling pathways in each lineage. Expression of Wnt regulated TF *Tcf7l1* along with Wnt receptor *Ror2* were maintained from LaTP through commitment to a SynTII precursor state. Upon commitment to the SynTII lineage, the TF *Zfhx3* and growth factor receptor *Igf1r* are newly expressed while canonical markers *Gcm1* and *Synb* are among the later transcript changes (*Figure 2C*). SynTI Precursor cells maintain *Pvt1* expression but uniquely upregulate the ephrin receptor *Epha4*, the Egf-related ligand *Tgfa*, and the Egf receptor substrate *Eps8*. Only mature SynTI express the kinase *Prkce* and clathrin coat assembly protein *Snap91* (*Figure 2D*). S-TGC Precursor cells activate the receptors *Pparg* and *Lifr* as well as the hypoxia induced TF *Epas1* upon lineage commitment. Expression of the canonical marker *Ctsq* occurs only in mature S-TGC along with *Lepr* and the glucocorticoid signaling effector *Tsc22d3* (*Figure 2E*). These data are consistent with distinct signaling pathways (e.g. SynTII – WNT, IGF; SynTI – EGF, EPH; S-TGC – LIF, PPAR) driving differentiation of LaTP into the three main cell types of the placental labyrinth. Interestingly, the S-TGC appear to differentiate from the junction of LaTP2 and JZP1 (*Figure 2A*), potentially suggestive that S-TGC arise from ectoplacental cone cells (*Simmons et al., 2007*; *Simmons et al., 2008a*) as well as from chorion derived LaTP (*Ueno et al., 2013*). It is also possible the location of the S-TGC near JZP1 reflects the shared prolactin expression between S-TGC and other junctional zone populations. Again, lineage tracing will be required to address their developmental relationships.

As described above, the remaining two arms were consistent with differentiation of the JZ populations: SpT and GC cells. Immunofluorescence of the representative markers NCAM1 and SLCO2A1 confirmed their expression in the JZ (*Figure 2F*). Furthermore, staining of juxtaposed sections confirmed their mutually exclusive expression (*Figure 2—figure supplement 2B*). Co-staining with the known GC marker PCDH12 (*Rampon et al., 2005*), verified the GC identity of the NCAM1+ cells (*Figure 2—figure supplement 2C*). Previous work has identified differential expression of prolactin genes between GC and SpT (*Simmons et al., 2008a*). Therefore, we analyzed the expression of these genes in our dataset. Expression analysis of prolactin genes showed both distinct and overlapping expression of this large gene family between the SpT and GC cells, and confirmed the identities of these clusters (*Figure 2—figure supplement 3A and B*). Of important note, parietal TGC nuclei were not identified in our snRNA-seq data, likely due to their very large size precluding their passage through the FACS pre-filter, which removes nuclei greater than 35 microns (*Figure 2—figure supplement 2D*). Our data also did not identify invasive trophoblast giant cells (spiral artery and canal TGCs) that invade the decidua, which was dissected away. Still, this analysis of the trophoblast snRNA-seq was able to clarify the cell types and their relationships for a predominance of the lineages of the mouse placenta identifying novel intermediate states and markers of the different cell populations.

## Developmental time course and trajectory inference reveal details of lineage dynamics and commitment

Next, we asked how the trophoblast cell populations changed over developmental time by separating the conglomerate data into its individual timepoints (*Figure 3A*). This visualization clearly showed the rapid diminishment of the LaTP populations over time, resulting in few progenitors remaining by E14.5 (*Figure 3A and B*, *Supplementary file 3*). The precursor populations also decreased over time, presumably maturing into terminally differentiated cells, which dramatically increased during the time course (*Figure 3A and B*). To determine which populations were primarily responsible for the expansion of the number of cells in the placenta, we evaluated expression of the cell cycle marker *Mki67* which is lost when cells exit the cell cycle (*Gerdes et al., 1984*). The majority of *Mki67* expressing nuclei were among the progenitor and precursor cell populations (*Figure 3—figure supplement 1A*). However, at E9.5 and E10.5, a substantial number of nuclei at the distal tips of each arm also expressed *Mki67*, suggesting ongoing proliferation even after cells have transitioned to a more mature-like expression state. Very few nuclei expressed *Mki67* at E14.5 consistent with the expansion in placental cell number happening prior to E14.5, with later growth occurring through cellular hypertrophy rather than cell division (*Ueno et al., 2013*; *Paikari et al., 2017*).

Cellular dynamics can be tracked not only by collecting samples at different developmental time points, but also by leveraging splicing information inferred through comparisons of intron vs. exonic

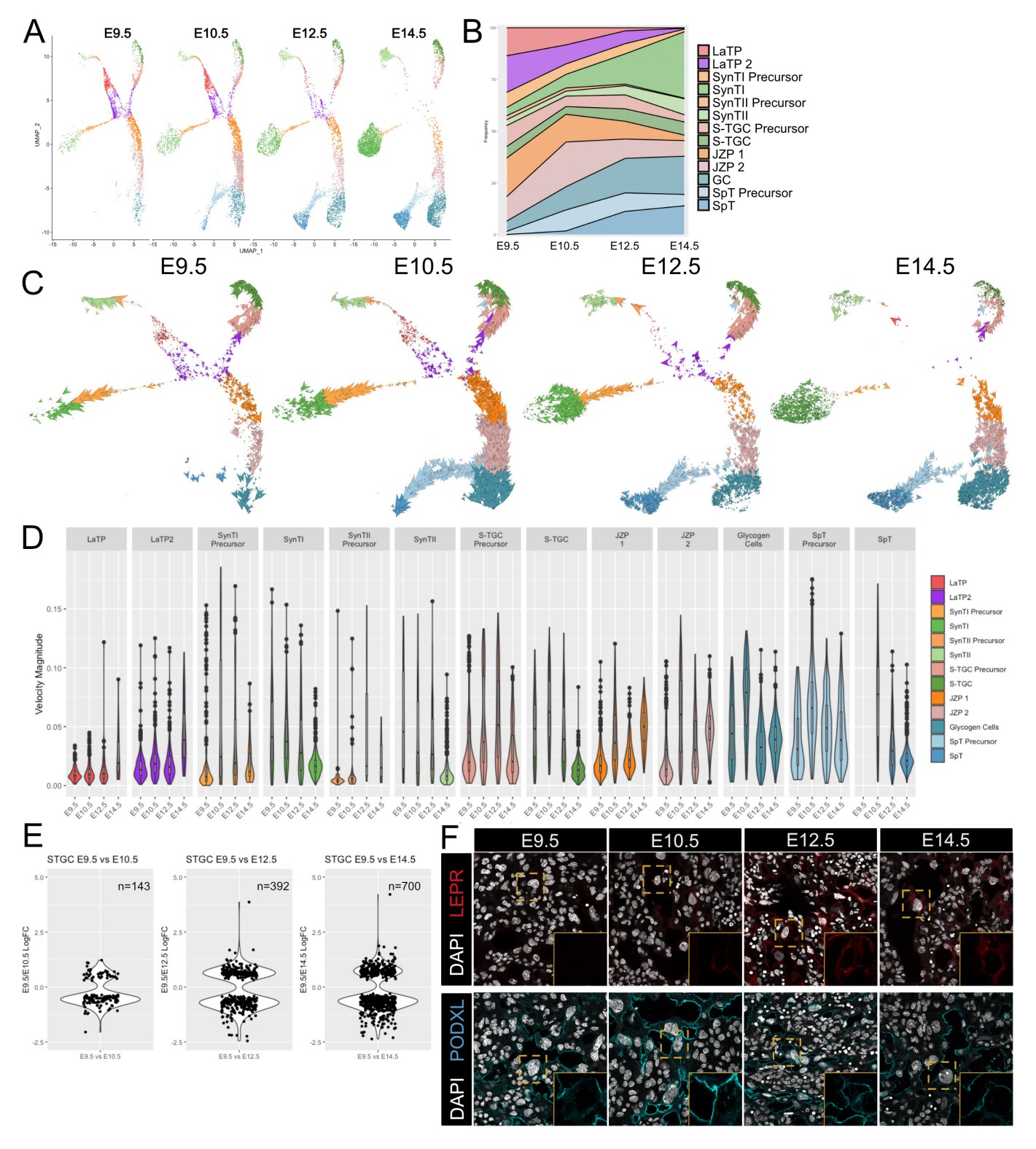

**Figure 3.** Developmental time course and trajectory inference reveal details of lineage dynamics and commitment in trophoblast. (**A**) UMAP projection of all nuclei captured at each gestational age. (**B**) Quantification of the proportion of each cluster captured at each developmental time point. (**C**) RNA velocity vectors of the trophoblast nuclei at each time point. Arrows show the estimated magnitude and direction of each nucleus in pseudotime. (**D**) The magnitude of velocity vectors shown in D summarized as violin plots, with nuclei split by cluster identity and developmental time point. (**E**) Violin plots showing the log2 fold change of differentially expressed genes between nuclei from S-TGC at E9.5 vs. each other timepoint. (**F**)

*Figure 3 continued on next page*

*Figure 3 continued*

Immunofluorescence staining of LEPR (top) and PODXL (bottom) in the placental labyrinth at each developmental timepoint at 63x. Insets, magnification of area surrounding a S-TGC indicated by dotted line, DAPI stain removed to highlight protein expression.

The online version of this article includes the following source data and figure supplement(s) for figure 3:

**Source data 1.** Average RNA velocity magnitude split by developmental stage and cluster (*Figure 3D*).
**Source data 2.** S-TGC Differential Expression (E9.5 v E10.5).
**Source data 3.** S-TGC Differential Expression (E9.5 v E12.5).
**Source data 4.** S-TGC Differential Expression (E9.5 v E14.5).
**Figure supplement 1.** Analysis of S-TGC across developmental time reveals a switch from proliferation to maturation.
**Figure supplement 1—source data 1.** S-TGC Gene Ontology results by developmental stage.

reads of genes within individual cells, termed RNA velocity. We performed RNA velocity using the scVelo package to produce vectors representing both the direction and speed of cellular maturation (*Bergen et al., 2019*). In the scVelo visualization, each nucleus is identified by an arrow representing the inferred direction and rate of cellular change projected in UMAP space. The data was visualized separately for each different developmental timepoint to discern the dynamics of cell transitions and how these dynamics change over time (*Figure 3C*). In addition, the velocity magnitude was summarized for each cell population at each developmental timepoint (*Figure 3D*, *Figure 3—source data 1*).

These analyses uncovered a number of interesting properties associated with placental development. Progenitor vectors (LaTP and LaTP2) showed little directionality and small magnitudes, likely reflecting a maintenance of multipotency and self-renewal. In contrast, the cells within each precursor population showed uniform directionality and large magnitudes, especially at E10.5 and E12.5, the period of greatest placental expansion. Surprisingly, the differentiated cell populations also showed vectors of great magnitude and uniform directionality at the early timepoints, suggesting ongoing transcriptional maturation, before diminishing by E14.5. To evaluate ongoing changes in the mature populations, we selected the S-TGC population and performed differential expressional analysis between each timepoint and E9.5. Consistent with the vectors, the number of differentially expressed genes at each consecutive timepoint increased from 143 to 700 genes (adj. p-value<0.05; *Figure 3E*, *Figure 3—figure supplement 1B*, *Figure 3—source datas 2–4*).

Gene ontology analysis showed that genes up at E9.5 were highly enriched for ontology groups concerning cell division, while those at E14.5 were enriched for Ras GTPase signaling and various developmental processes (*Figure 3—figure supplement 1C*, *Figure 3—figure supplement 1—source data 1*). Examples of genes that increased over developmental time include *Ctsq*, the Cathepsins (*Ctsj*, *Ctsr*, *Ctsm*, *Cts3*, and *Cts6*), and several genes involved in hormone signaling and response (*Lepr*, *Tsc22d3*, and *Ghrh*) (*Figure 3—figure supplement 1B,D,E,F*). In contrast, *Podxl* is expressed from the very early stages of S-TGC specification through E14.5. *Podxl* is a highly charged membrane protein which, in a different context, has been implicated in luminogenesis in early embryos (*Shahbazi et al., 2017*). While the formation of maternal blood spaces in the placental is distinct from the early embryo, the early expression of Podxl in S-TGCs might function in maintaining open maternal blood spaces within the placental labyrinth. The differential dynamics of PODXL and LEPR in S-TGCs was confirmed by immunohistochemistry (*Figure 3F*). Similar ongoing maturation processes were seen in the SynTII, SynTI, GC, and SpT populations (*Figure 3—figure supplement 1G*). Together these data support a model of early expansion by self-renewing progenitor cells and committed precursors followed by ongoing maturation of all five differentiated cell types of the placenta proper.

## Defining distinct roles of the trophoblast subtypes at the gas exchange interface

The labyrinth is the site of transport between fetal and maternal blood in the mouse placenta. Fetal and maternal blood are separated by a thin membrane consisting of four cell layers starting with the fetal endothelial cells, then SynTII, followed by SynTI, and finally S-TGCs (*Simmons et al., 2008a*; *Figure 4A*). A basal lamina separates the fetal endothelial cells from SynTII cells. The expression programs and functions of the three trophoblast cell layers have been difficult to dissect due to

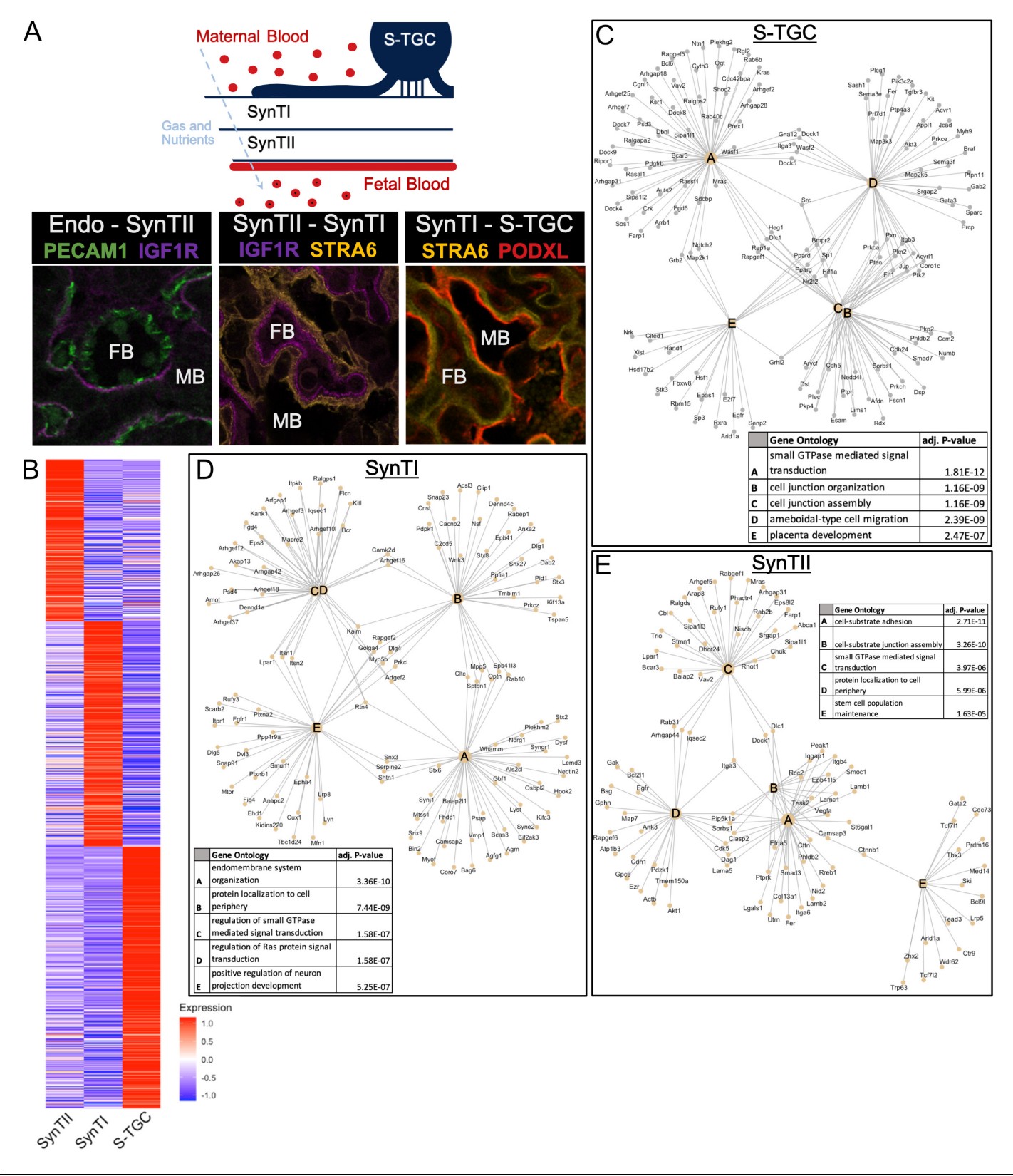

**Figure 4.** Defining distinct roles of the trophoblast subtypes at the gas exchange interface. (**A**) Schematic showing the relative location of the cell types at the gas and nutrient exchange interface of the labyrinth – S-TGC, SynTI, SynTII, (sometimes referred to as trophoblast layer I, II, and III, respectively)

*Figure 4 continued on next page*

eLife Research article

Developmental Biology

*Figure 4 continued*

and the fetal endothelium (top). Immunofluorescence staining resolving expression in each pair of neighboring cell types – Endothelial/SynTII, SynTII/ SynTI, and SynTI/S-TGC in E12.5 mouse placenta sections. (B) Heatmap of all differentially expressed genes by log fold change in SynTI, SynTII, and S-TGC clusters. (C–E) Network diagrams showing the top 5 Gene Ontology (Biological Process) enriched in marker genes (adj. p-value<0.05, logFC >0.5) in each cluster (Yu G. 2018). Gene ontology categories are nodes and associated marker genes are annotated. Adjusted p-values for enrichment are listed in the table insets.

The online version of this article includes the following source data and figure supplement(s) for figure 4:

**Source data 1.** Differential expression between interface clusters (SynTI, SynTII, and S-TGC).
**Source data 2.** Gene Ontology results for differentially expressed genes between interface clusters (SynTI, SynTII, and S-TGC).
**Figure supplement 1.** Gene ontology (biological process) of differentially expressed genes SynTII (top), SynTI (middle), S-TGC (bottom).

the syncytial nature of the SynTI and SynTII cells. Isolation of nuclei followed by snRNA-seq overcomes this barrier enabling the determination of the unique expression programs of each trophoblast layer. Immunohistochemistry for the marker genes PECAM1, IGF1R, STRA6, and PODXL demonstrate each cluster corresponds to endothelial, SynTII, SynTI, and S-TGC populations respectively in vivo (*Figure 4A*). Each layer could be distinguished by high- resolution image analysis with nucleated erythrocytes residing in the PECAM1+ fetal vessels and large polyploid S-TGC residing in the maternal blood sinuses. PODXL+ cytoplasmic extensions of the S-TGCs spread over the perimeter of the maternal blood space forming a permeable lining (*Figure 4A*).

SynTII, SynTI, and S-TGCs were distinguished from one another based on their transcript profiles (*Figure 4B*, *Figure 4—source data 1*). Furthermore, gene ontology analysis of differentially expressed genes between populations suggested highly distinct functions (*Figure 4C–E* and *Figure 4—figure supplement 1*, *Figure 4—source data 2*). SynTII differentially expressed genes were highly enriched for cell–cell and cell–matrix interactions such as those encoding cell junction and cell adherens proteins consistent with a role as a basal permeability barrier between maternal and fetal blood. SynTII also expressed collagens and laminin subunits (*Figure 4E*), which likely form the basal lamina separating SynTII from the fetal endothelium. Also identified in SynTII differentially expressed genes was the stem cell population maintenance GO category, which included several genes known to be important for placental function though not specific to stem cells (Bcl9, Gata2, and Tbx3).

In contrast, SynTI differentially expressed genes were enriched for intracellular transport and cell signaling including genes encoding for vesicular organization and transport, protein localization to membranes, and Ras signaling. These data are consistent with SynTI layer playing a critical role in the transport of materials between maternal and fetal blood. SynTI is also enriched for genes involved in neuron projection development, driven by expression of canonical axonal pathfinding genes (Plxna2, Plxnd1) and synapse genes (Snap91). The exact function of these genes in SynTI is unclear and would be of interest for future study.

Although more generally understood as an exocrine cell type, the S-TGC were enriched for GTPase and Ras signaling and receptors including Kras, Grb2, and Pdgfrb, suggesting these cells also transduce signals from maternal blood. S-TGC differentially expressed genes were strikingly enriched for those encoding for migration factors. Such expression could explain the long thin cellular extensions from these cells, which form the only non-syncytial trophoblast at the interface (*Coan et al., 2005*). S-TGCs are connected to SynTI by desmosomes, which are known to undergo constant remodeling and have additional functions in signaling and migration (*Simmons et al., 2008a*; *Johnson et al., 2014*). These S-TGC-specific programs along with the high expression of *Podxl*, likely underlie a central role for these cells in the function of the maternal sinuses. S-TGC are also enriched for the placental development category which includes several genes important for giant cell function including *Hand1* (regulation of giant cell lineage), *Hsd17b2* (steroid hormone production), and *E2F7* (endoreduplication).

Given that the labyrinth represents the site of exchange of nutrients, minerals, metabolites, ions, gases, and regulatory factors, between the mother and fetus, we next wanted to better understand how these functions are distributed across the different trophoblast layers. We focused on two large functional categories: solute transporter and cholesterol/vitamin transport. All genes in these categories that were found to be differentially expressed between one of the interface populations (SynTI, SynTII, S-TGC) compared to all other trophoblast populations were included in this analysis. The results suggested a striking separation of functions between the layers (*Figure 4—figure*

*supplement 1B and C*). For example, genes responsible for carboxylate transport were highly enriched in SynTII, while organic anion transporters were enriched in SynTI. Even within specific transport pathways, functions were often separated; for example, factors important in Zinc transport were distributed across, yet distinctly expressed between the populations. Cholesterol transporters are enriched in SynTII (*Scarb1*, *Vldlr*) and S-TGC (*Hdlbp*, *Ldlr*), while Folate receptors (*Folr1*, *Folr2*) are uniquely expressed in SynTI. Together, these data show a highly evolved separation of functions between the three trophoblast layers that separate maternal from fetal blood.

## Predicting cell signaling within the placental labyrinth

Single-cell RNA-seq has been used to predict signaling between different cell types of a tissue or organ (*Efremova et al., 2020*; *Vento-Tormo et al., 2018*). Here, we asked whether we could do the same with snRNA-seq to provide insight into the developmental signals driving mouse placental development. We used the package CellPhoneDB (*Efremova et al., 2020*) which applies a data base of annotated receptor-ligand pairs to the single-cell expression data evaluating all possible combinations of cell types (i.e. identified clusters). We focused our analysis on the populations of the developing labyrinth (LaTP, LaTP2, SynTI Precursor, SynTI, SynTII, S-TGC Precursor, S-TGC, and fetal endothelium). Given that CellPhoneDB receptor-ligand database was developed for human cells, we only included orthologous genes from our mouse snRNA-seq data. The results are summarized as a heatmap of the predicted strength (average log 2 expression of ligand-receptor pair) in *Figure 5—figure supplement 1* with receptor–ligand interactions shown on the rows and each cluster pair shown as columns (full data in *Figure 5—figure supplement 1—source data 1*). A total of 139 unique ligand–receptor interactions reaching significance (adj. p-value<0.05) were uncovered.

CellPhoneDB predicted an interaction between *Rspo3* in the endothelium with *Lgr5* in the LaTP/SynTII lineage (*Figure 5A*). *Rspo3* is required in vivo for the maintenance of *Gcm1* expression (*Kazanskaya et al., 2008*; *Aoki et al., 2007*) and canonical Wnt signaling is essential for differentiation of SynTII (*Zhu et al., 2017*; *Matsuura et al., 2011*). Predicted signaling in the opposite direction included *Vegfa* secreted by SynTII to the receptors *Kdr*, *Flt1*, *Nrp1*, *Nrp2* located only on endothelial cells. Immunostaining confirmed SynTII and SynTI as the sources of VEGFA in the labyrinth (*Figure 5B*, *Figure 5—figure supplement 5–S2A*). These data suggest a feedback loop where *Rspo3-Lgr5* supports SynTII differentiation and function, while *Vegfa-Kdr/Flt1* drives vascular maintenance and remodeling, thereby coordinating lineage differentiation with development of tissue structure.

Analysis of marker genes suggested Egf signaling as a regulator of LaTP differentiation to SynTI (*Figure 2D*). CellPhoneDB analysis was consistent with that interpretation as it predicted an interaction between the EGFR ligands *Tgfa*, *Hbegf,* and *Nrg1* produced by SynTI with *Egfr* on both LaTP2 and SynTI precursor populations. However, it also predicted an interaction with *Egfr* on LaTP, SynTII, and S-TGCs (*Figure 5C*). We confirmed protein expression HBEGF alone in SynTI, and colocalization of HBEGF and EGFR in SynTII (*Figure 5D*; *Figure 5—figure supplement 2B and C*). Therefore, SynTI appears to be a hub of EGF signaling that impacts the majority of the cell populations in the labyrinth. Loss of *Egfr* in vivo results in decreased placental size, disorganized labyrinth, and a reduction in the SpT layer (*Du et al., 2004*; *Miettinen et al., 1995*; *Strunk et al., 2004*). Depletion of Egf ligands yields a variety of phenotypes (*Liu et al., 2019*; *Luetteke et al., 1993*). Taken together, these data suggest a complex regulation of Egfr signaling being coordinated by SynTI that is critical for proper development of the labyrinth.

Numerous ligand–receptor interactions between *Bmp8a* and several receptors, including *Acvr1*, *Bmpr1*, and *Bmpr2* were predicted in the labyrinth. The source of *Bmp8a* is LaTP and SynTII and the receptors are located on all labyrinth populations, including LaTP and SynTII (*Figure 5E*). Additional Bmp 60A subfamily members, *Bmp6* and *Bmp7*, were highly secreted. *Bmp7* is secreted by LaTP and *Bmp6* by the endothelium (*Figure 5E and F*). In addition to BMP ligand and receptors, the pathway's downstream effectors, the *Smad* TFs, also showed striking differential expression between the cell types of the labyrinth. *Smad2/4/7* were relatively uniformly expressed in trophoblasts, while *Smad1/3/5/6* displayed varied levels of expression across trophoblast populations (*Figure 5G* and *Figure 5—figure supplement 2D*). Previous work has shown that while *Bmp8a* is largely redundant with *Bmp8b*, loss of *Smad1*, and combined knockout of *Bmp5/7* exhibit severe placental defects (*Zhao and Hogan, 1996*; *Tremblay et al., 2001*; *Solloway and Robertson, 1999*). Cell-type-specific perturbation of elements of the Bmp pathway has not been performed; however, ligand–receptor

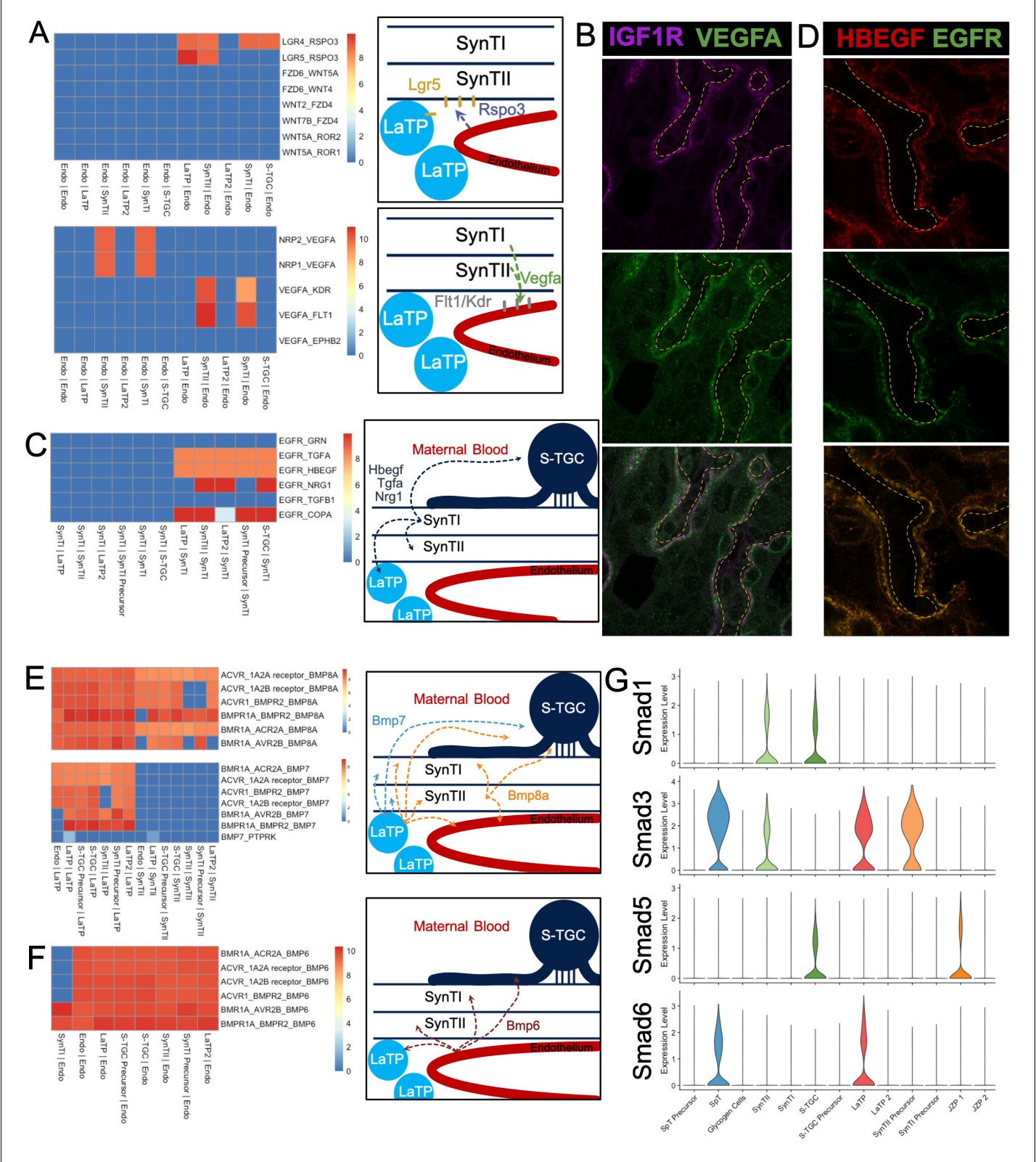

**Figure 5.** Predicting Cell Signaling within the placental labyrinth. (**A**) Heatmap showing all Wnt signaling interactions (top) and Vegf signaling (bottom) between the endothelium and labyrinth trophoblast populations. Left: schematics summarizing signaling interactions. (**B**) Immunofluorescence staining for VEGFA and IGF1R demonstrating colocalization in SynTII in E12.5 mouse placenta sections (63x). Fetal Vessels are outlined in yellow dashed line. (**C**) Heatmap showing all Egfr signaling interactions between labyrinth populations. Schematic showing secretion of Egf ligands *Hbegf*, *Tgfa*, and *Nrg1*

*Figure 5 continued on next page*

*Figure 5 continued*

from SynTI to *Egfr* expressed on SynTII, LaTP, and S-TGC. (**D**) Immunofluorescence staining for HBEGF and EGFR demonstrating colocalization of the ligand and receptor SynTII, but only HBEGF expression in SynTI (E12.5 mouse placenta sections - 63x) (**E**) Heatmap showing signaling interactions of *Bmp8a* (top) and *Bmp7* (bottom), secreted by the LaTP and SynTII or LaTP only, respectively. (**F**) Heatmap showing predicted signaling interactions between *Bmp6* secreted by the endothelium to receptors on LaTP, SynTI, SynTII and S-TGC. (**G**) Transcript expression for the *Smad* TFs which are differentially regulated (*Smad1/3/5/6*).

The online version of this article includes the following source data and figure supplement(s) for figure 5:

**Figure supplement 1.** All predicted receptor-ligand interactions between labyrinth populations.
**Figure supplement 1—source data 1.** CellPhoneDB interaction raw data for trophoblast populations.
**Figure supplement 2.** Validation of predicted cell signaling events in the labyrinth.

modeling with CellPhoneDB combined with snRNA-seq expression data provides insight into how the pathway likely functions in vivo.

## Modeling transcription factor regulon activity identifies new candidate regulators of SynTII

Transcription factors (TFs) are the most direct regulators of cell fate and can be used to reprogram cells to new cell fates (*Kubaczka et al., 2015*). Our snRNA-seq data contained many well-studied TFs, but expression alone is not necessarily a good indicator of function. TFs typically act cooperatively with other TFs and numerous factors influence their activity. Therefore, we sought to identify active TFs based on both expression of the TF and its downstream targets within the same cells. To do so, we used the SCENIC package (*Aibar et al., 2017*) which uses a database of TFs and their annotated motifs to identify enrichment of a TF motif within or near the promoter regions of co-expressed genes within each cell to define 'regulon activity'. As such SCENIC infers activity of a transcriptional network rather than considering expression of the TF alone.

Application of SCENIC to the trophoblast dataset recovered 200 transcription factor regulons with activity in at least one cluster (*Figure 6—source data 1*). To ensure that the regulon activity data maintained the variation of the transcript expression data, we projected the nuclei in UMAP space using regulon activity transcript expression as the underlying variable. The resulting plot was color coded based on a cluster identity as defined by Seurat analysis of transcript expression (*Figure 6A*). There was overall strong agreement in the two approaches to separate nuclei by cell type both in UMAP space and hierarchical clustering of Pearson correlations between populations (*Figure 2A* vs. *Figure 6A*, *Figure 6—figure supplement 1A*). These data show that TF regulon activity provides an alternative means to identify the different cell types of the placenta.

The identified regulons with differential activity across the populations included both known and unknown TF regulators of placental development. Known factors include *Gcm1* in SynTII specification, *Grhl2* in S-TGC, and *Ets2* in JZ development (*Figure 6B*; *Anson-Cartwright et al., 2000*; *Walentin et al., 2015*; *Yamamoto et al., 1998*). Examples of the many unknowns include *Tbx15* in LaTP, *Gata1* in SynTII, *Pax2* in SynTI, and *Meis1* in S-TGC. The power of the regulon approach is further supported by comparing the distribution of TF expression alone versus that of their regulon (*Figure 6C*). While expression of *Gcm1* and *Pax8* correlated the distribution of their regulon activity, *Ets2* and *Esrrg* did not. *Ets2* was broadly expressed, yet its regulon activity was highly enriched among differentiating S-TGCs and GCs. *Esrrg*, on the other hand, was lowly expressed throughout, yet showed strong regulon active among SpT cells. *Esrrg* is not well studied in the mouse placenta, but has known roles in steroid hormone production and metabolism in human trophoblast, functions also important in the mouse placenta (*Poidatz et al., 2012*; *Luo et al., 2014*).

Given that most of the TFs underlying cluster-specific regulons have not been studied in the placenta, we next researched the Mouse Genome Informatic database for any phenotypes associated with the knockout of TFs driving the uncovered regulons. Only 17 of the 200 TFs had annotated placental phenotypes, while 124 are lethal during development (defined as until weaning) (*Figure 6D*). Many of these developmental lethal phenotypes likely are associated with placental defects given that 68% of 103 intrauterine lethal and subviable mouse knockout lines also have placental defects (*Perez-Garcia et al., 2018*). Unfortunately, few of knockouts evaluated in the Perez-Garcia et al paper were for transcription factors, making comparisons to our regulon data impossible. However,

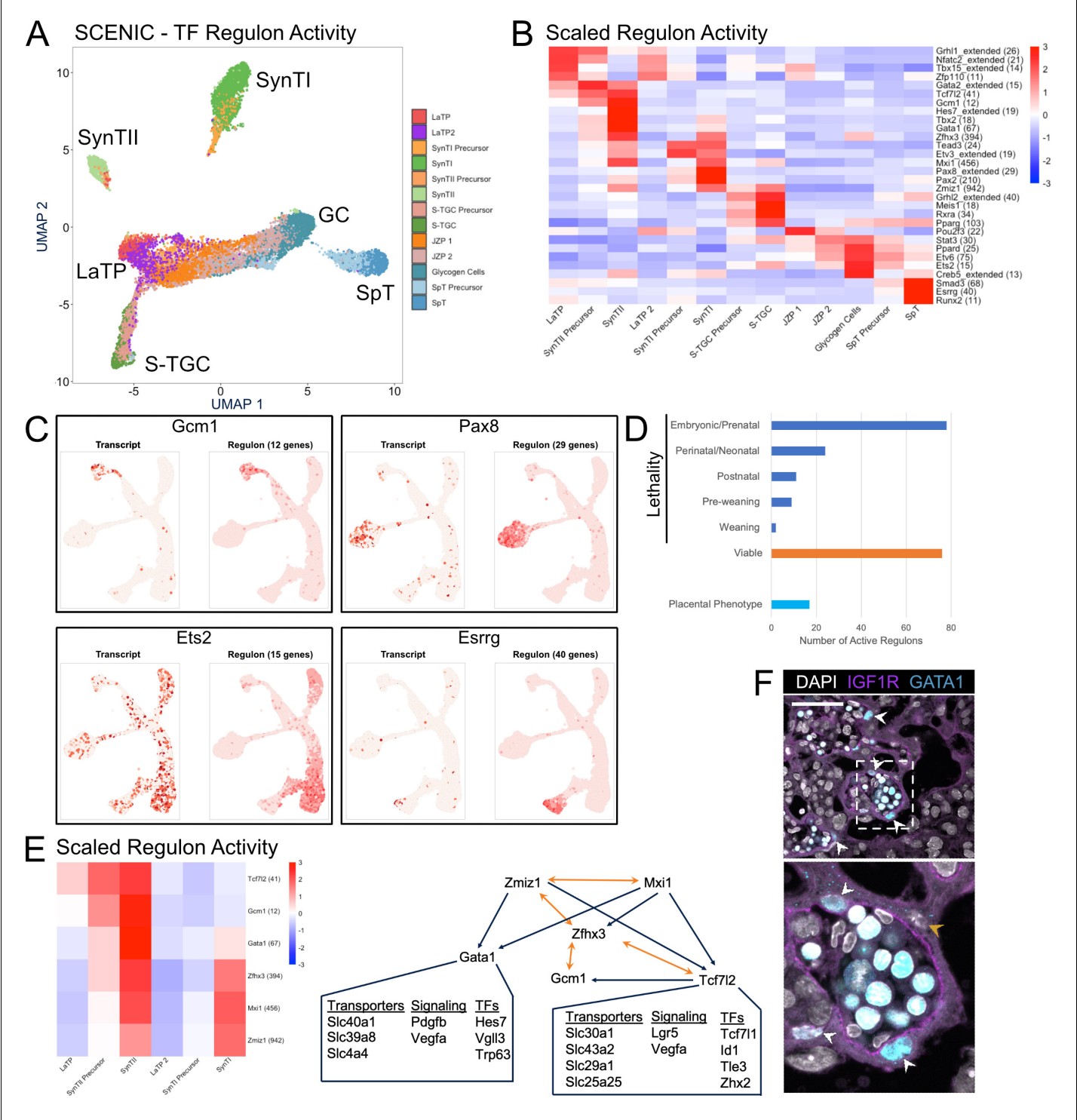

**Figure 6.** Modeling transcription factor regulon activity identifies new candidate regulators of SynTII. (A) UMAP projection derived from regulon activity predicted by SCENIC. The clusters are colored according to Seurat clustering of transcript data. The cluster identities of each of the five arms of the UMAP are annotated. (B) Heatmap of the scaled regulon activity for select regulons expressed in each trophoblast population. The number of genes predicted to be regulated by each transcription factor is included in parenthesis. (C) Comparison of transcript expression of the TF (left) and the regulon activity for that TF (right) at the single-cell resolution projected in UMAP space for *Gcm1*, *Pax8*, *Ets2*, and *Esrrg*. The number of genes predicted to be regulated by each transcription factor is denoted in parentheses. (D) Quantification of the number active regulons with a lethal phenotype in mouse according to the Jackson Laboratory MGI (Embryonic/Prenatal - 78, Perinatal/Neonatal - 24, Postnatal - 11, Pre-weaning - 9, lethality at weaning - 2, no developmental lethality - 76) and the number with known placental phenotypes in vivo (n = 17). (E) Heatmap showing

*Figure 6 continued on next page*

*Figure 6 continued*

expression of a network of regulons enriched for expression in SynTII (left). Gene regulatory network of high confidence predicted interactions from GRNboost2 (right). Unidirectional regulation denoted by blue arrows and bidirectional regulation by yellow. Predicted targets of *Gata1* and *Tcf7l2* expressed in SynTII and important in SynTII differentiation or function are annotated. (**F**) Immunofluorescence staining for GATA1 and IGF1R showing GATA1 expressing nuclei in IGF1R SynTII cells (E12.5 mouse placenta sections - 63x). GATA1+ SynTII nuclei – white arrowheads, GATA1- putative SynTI nucleus – yellow arrowhead.

The online version of this article includes the following source data and figure supplement(s) for figure 6:

**Source data 1.** SCENIC scaled regulon activity in trophoblast nuclei.
**Source data 2.** SCENIC regulon predicted binding targets for all active regulons.
**Figure supplement 1.** Additional validation of SCENIC and Gata1 expression and expression of genes with validated placental phenotypes in vivo.

we did see robust and cell-type-specific expression of several factors uncovered in their paper as having placental phenotypes, including *Fryl* in GCs, *Atp11a* in the SynTI lineage, and *Adcy9* in SynTII (*Figure 6—figure supplement 1B and C*). Together, these analyses suggest likely important roles for the uncovered regulons in placental development.

Next, we asked if we could uncover networks among the TFs themselves focusing on the SynTII lineage. In particular, the regulon data was evaluated for evidence of coexistence of paired regulon activity with associated high confidence annotations (defined by either direct annotation or inferred by orthology) of binding sites for one or both of the TFs in the other TF's promoter region (*Figure 6E*, *Figure 6—source data 2*). *Gcm1*, the master regulator of the SynTI lineage, and *Zhfx3* have reciprocal high confidence motif in their promoter regions, suggestive of a positive feedback loop. *Zfhx3* and *Tcf7l2* also show mutual regulation of one another, and *Tcf7l2* also has a high confidence binding site in the promoter of *Gcm1*. Thus, these 3 TFs establish what appears to be an autoregulatory loop. *Zfhx3* (*Atbf1*) does not have known functions in the placenta. However, two independent genetic disruptions of *Zfhx3* result in embryonic to pre-weaning lethality with surviving pups exhibiting severe growth defects (*Sun et al., 2012*; *Parsons et al., 2015*). While *Tcf7l2* knockout mice do not have a placental phenotype, we identify high confidence motifs in the promoter regions of many genes important for SynTII differentiation and function including *Lgr5*, *Vegfa*, *Slc43a2*, and *Gcm1* (*Perez-Garcia et al., 2018*; *Kazanskaya et al., 2008*; *Guetg et al., 2015*). It is possible that redundancy with *Tcf7l1* underlies the lack of *Tcf7l2* knockout placental phenotype (*Nguyen et al., 2009*).

*Zfhx3* also appears to be in a positive regulatory loop with the two TFs *Zmiz1* and *Mxi1*. *Zmiz1* and *Mxi1* in turn both have high confidence binding sites in the promoter of *Gata1*. While *Gata1* has not been studied in the context of the mouse placenta, transcript expression, regulon activity, and high confidence motifs suggest it regulates multiple genes with important functions in SynTII including *Vegfa*, *Slc40a1* (ferroportin), and the basal epithelial transcription factor *Trp63*. Immunofluorescence showed the colocalization of GATA1 with IGFR1, thereby confirming its expression in SynTII (*Figure 6F*). In particular, strong GATA1 protein expression was visible in the SynTII nuclei abutting the basal membrane (*Figure 6F* and *Figure 6—figure supplement 1D* – white arrow heads), but not in SynTI nuclei closest to maternal blood spaces (*Figure 6F* – yellow arrowhead). Strong GATA1 staining was also obvious in the nuclei of fetal erythrocytes with vascular lumen. Therefore, GATA1 appears to be expressed exclusively in SynTII population of trophoblasts and likely has important regulatory functions specific to SynTII.

## Discussion

This study profiles the development of the mouse placenta by transcriptome capture of nuclei, rather than whole cells, resulting in a high-resolution map of labyrinth differentiation and allowing for the first independent analyses of the two SynT populations. In contrast to previous studies where SynT comprise fewer than 10% of cells identified, almost 30% of trophoblast nuclei captured in this study originate from SynT cells. Profiling four developmental stages from E9.5 to E14.5 allowed for the capture of stem/progenitor and terminally differentiated populations including previously unknown intermediate states and new putative regulators of cell identity. Modeling receptor–ligand interactions predicted how SynTII and endothelial cells coordinate spatial proximity and development through growth factor secretion. In addition, network inference predicted gene regulatory networks

defined by TFs and feedback loops among the TFs, including novel candidate regulators of placental development. All together, these data represent a high-resolution map of placental labyrinth development providing insights into the cell autonomous and non-autonomous molecular events underlying the formation and function of this remarkable organ.

Understanding the development of any tissue begins with the progenitor populations. Previous work uncovered a population of EPCAM and MET expressing progenitors that give rise to the three trophoblast populations of the labyrinth (SynTI, SynTII, and S-TGC) (*Ueno et al., 2013*). Clustering analysis of the snRNA-seq data separated these cells into two closely related clusters, LaTP and LaTP2. Both expressed *Epcam*, but only LaTP expressed *Met*. The relationship between these populations is unclear. Both LaTP populations had low RNA velocities and there was no coordinated directionality to indicate differentiation from one population to the other. However, the relative positions of the two populations in the developing labyrinth and the earlier presence of EPCAM/MET double positive cells (i.e. at E8.5) suggest that LaTP2 arise from LaTP. Alternatively, the two populations may arise from progenitors that separate earlier in development resulting in distinct lineage biases. LaTP is more similar to SynTII, while LaTP2 is more similar to SynTI and S-TGC. Understanding the specification and heterogeneity of labyrinth progenitors will be an important undertaking given that early defects are likely to have large effects in proportion of terminally differentiated trophoblast resulting in embryonic defects and ultimately reduced viability.

Tracking differentiation from the LaTP populations identified previously unknown precursor populations along differentiation to SynTI and S-TGC. Profiling differentiation to SynTII confirmed previous work demonstrating the role of Wnt signaling in specification and efficient differentiation of SynTII, both in vivo and in vitro (*Zhu et al., 2017*; *Matsuura et al., 2011*). However, such detailed information and derivation protocols did not exist for SynTI or S-TGC. SynTI precursor cells strongly upregulated Egf signaling members *Tgfa* and *Eps8*. Additionally, expression of Egf ligands by mature SynTI and *Egfr* by LaTP 2, SynTII, and S-TGC underscore the centrality of Egf signaling to labyrinth development and function. While Egfr loss or mutation in vivo causes labyrinth disorganization, the deciphering of mechanism has been confounded by defects in junctional zone cells and strain-specific variation (*Du et al., 2004*; *Strunk et al., 2004*; *Dackor et al., 2009*; *Lee and Threadgill, 2009*). In human trophoblast, EGF has been shown to stimulate differentiation to SynT in vitro and decreased EGFR phosphorylation is associated with intrauterine growth restriction (*Morrish et al., 1987*; *Fondacci et al., 1994*). Given the redundancy of Egf ligands, a careful temporal and cell-type-specific dissection of the loss of *Egfr* will be necessary to separate the multiple functions of Egf signaling in placental development and health.

Pseudotemporal ordering along the S-TGC trajectory implicated Lif-Stat3 signaling evidenced by increasing expression of *Lifr* and supported by high activity of the predicted *Stat3* regulon by SCENIC. Loss of *Lifr* results in disorganization of the labyrinth and dramatically enlarged non-functional maternal blood sinuses lined by multinuclear aggregates, and not S-TGC (*Ware et al., 1995*). While Lif-Stat3 is also active in the junctional zone, this striking labyrinth phenotype is likely explained by a necessity for Lif signaling in the differentiation, patterning, and/or function of S-TGC. Deciphering how *Lifr* is necessary for S-TGC and the regulation of maternal blood spaces will require genetic manipulation specific to S-TGC. This will be important to understand establishment and maintenance of functional maternal blood spaces which are at the core of common placental diseases like preeclampsia, fetal growth restriction, and diabetic complications (*Ware et al., 1995*; *Roberts and Escudero, 2012*).

This study represents the first independent transcript profiling of each SynT layer in vivo. This was achieved by the isolation of nuclei prior to transcriptome capture. These data demonstrate the utility of snRNA-seq in profiling syncytial placental cells without damaging and lengthy enzymatic digestion or single-cell dissociation. Mouse is the most common model for human placentation, even though the species differ in organ morphology, especially with respect to the gas exchange interface. Human placentas have villi that float in a pool of maternal blood. The villi are covered by a single syncytial layer and have no clear S-TGC equivalent. Comparing the transcript profiles of the interface populations in mouse and human will help to understand how the functions of SynTI, SynTII, and S-TGC are distributed across the fewer trophoblast populations of the human placenta. Such information is also likely to provide important insight into how the different structures arose during evolution and which human developmental processes can be effectively modeled in mouse.

In our study of the mouse labyrinth, specific marker genes identified in each layer provide insight into the functions of SynTI versus SynTII. Gene Ontology analysis of SynTI indicates enrichment for membrane associated proteins, including numerous transporters, receptors, and GTPase signaling factors. Many of these proteins, like STRA6, are localized to the apical surface where they transduce signals and uptake nutrients from maternal blood. However, SynTII also expressed many classes of solute transporters and signaling receptors, reflecting the export of molecules to the fetal endothelium. SynT layer-specific expression indicates which transporters are likely involved in uptake from maternal blood (SynTI enriched), export to fetal blood (SynTII enriched), or both (SynTI and SynTII enriched). Transcript data represent the first step in understanding the chain of nutrient transport and inform the critical cell type for targeted therapies for nutrient deficiencies.

Surprisingly, SynTII display unique enrichment for adhesion and junctional proteins, likely necessary to support frequent gap junctions on the border with SynTI and robust attachment to the basal lamina, respectively. Notably, several laminin, integrin, and collagen subunits are highly expressed. Electron microscopy has demonstrated only one basal lamina exists, which is between SynTII and fetal endothelial cells (*Coan et al., 2005*). Endothelial cells express a complementary set of extracellular matrix protein subunits (*Col4a1*, *Col5a2*, *Col18a1*), suggesting deposition of this membrane is a coordinated process between these two cell types. Coordinated development between SynTII and the endothelium is also found in a positive feedback loop of necessary growth factors (Wnt and Vegf). These data support an additional role for SynTII as a vascular support cell and likely cell type of origin for many placental vascular abnormalities.

Recent work identified that a surprising proportion of developmental lethal mouse mutants had unidentified placental phenotypes (*Perez-Garcia et al., 2018*). At least 200 TF defined regulatory networks are active in trophoblast during development. Over 62% of these TFs exhibit developmental lethal phenotypes upon homozygous loss of function but only 8.5% have published placental phenotypes (according to MGI). Candidate regulators were identified in all cell types, including *Zfp110* in LaTP, *Pax2* in SynTI, *Gata1* in SynTII, *Meis1* in S-TGC, and *Etv6* in GCs. For some TFs with strong phenotypes in other organ systems, a methodical analysis of placental morphology may not have been performed or published. For example, *Zfhx3* is predicted to form an autoregulatory loop with *Gcm1* and regulate several genes important to SynTII function. While loss of *Zfhx3* results in developmental lethality and severe growth defects, its role in placental development has not been examined (*Sun et al., 2012*; *Parsons et al., 2015*). Several TFs may have not been identified due to overlapping or compensatory roles with other proteins. One such example is *Tcf7l1* and *Tcf7l2*. While highly active in the SynTII, *Tcf7l2* exhibits no placental abnormalities upon knockout. *Tcf7l1*, which is also highly active in LaTP, has demonstrated redundancy with *Tcf7l2* in the skin and it is likely each factor compensates for loss of the other (*Nguyen et al., 2009*). As the knockout of both factors is lethal prior to the genesis of the labyrinth, conditional deletion in only extraembryonic tissues would be required to investigate their function in the placenta. While conditional or temporal control over deletion may be necessary for other factors inducing early lethality (*Klf5*) or confounding defects in other organ systems (*Gata1*), genetic deletions for almost all identified TFs exist and represent a large resource for immediate study of their effects in placental development.

In summary, this study represents a compendium of mouse placental labyrinth development from E9.5 through E14.5. These data provide a lens through which to better understand congenital defects, environmental and nutrient deficiencies, and species-specific differences in development and function.

## Materials and methods

### Key resources table

| Reagent type (species) or resource | Designation | Source or reference | Identifiers | Additional information |
|---|---|---|---|---|
| Strain (*Mus musculus*) | C57BL/6J Mice | https://www.jax.org/strain/000664 | 000664 | 6–12 weeks old |
| Commercial assay or kit | Nuclei isolation Kit: Nuclei EZ Prep | Sigma-Aldrich | NUC101-1KT | |

*Continued on next page*

*Continued*

| Reagent type (species) or resource | Designation | Source or reference | Identifiers | Additional information |
|---|---|---|---|---|
| Antibody | E-cadherin Monoclonal antibody (ECCD-2) | Thermofisher Scientific | 13–1900 | IF (1:250) |
| Antibody | Human/mouse NCAM-1/CD56 Polyclonal antibody | R and D Systems | AF2408-SP | IF (1:25) |
| Antibody | Anti-EpCAM Polyclonal antibody | Abcam | ab71916 | IF (1:1000) |
| Antibody | Anti-Stra6 Polyclonal antibody | Sigma-Aldrich | ABN1662 | IF (1:100) |
| Antibody | Anti-Igf1r Polyclonal antibody | R and D Systems | AF305 | IF (1:50 w/ antigen retrieval) |
| Antibody | Anti-Slco2a1 Polyclonal antibody | Atlas Antibodies | HPA013742 | IF (1:25 w/ antigen retrieval) |
| Antibody | Anti-Lepr Polyclonal antibody | R and D Systems | AF497 | IF (1:200) |
| Antibody | Anti-Pcdh12 Polyclonal antibody | Abcam | ab113720 | IF (1:25 w/ antigen retrieval) |
| Antibody | Anti-Podxl Polyclonal antibody | R and D Systems | AF1556 | IF (1:25) |
| Antibody | Anti-Pecam1 Polyclonal antibody | Abcam | ab23864 | IF (1:50) |
| Antibody | Anti-Vegfa Polyclonal antibody | Abcam | ab51745 | IF (1:50) |
| Antibody | Anti-Met Polyclonal antibody | R and D Systems | AF276 | IF (1:100) |
| Antibody | Anti-Egfr Polyclonal antibody | R and D Systems | AF1280 | IF (1:100) |
| Antibody | Anti-Gata1 Monoclonal antibody | Cell Signaling | 3535 | IF (1:100) |
| Antibody | Anti-Hbegf Polyclonal antibody | R and D Systems | AF8239 | IF (1:20) |
| Software, algorithm | R | https://www.r-project.org/ | | |
| Software, algorithm | ImageJ | ImageJ (http://imagej.nih.gov/ij/) | | |
| Software, algorithm | Seurat (3.1.3) | https://satijalab.org/seurat/ | | |
| Software, algorithm | cellranger (3.0.2) | https://support.10x genomics.com/single-cell-gene-expression/software/pipelines/latest/feature-bc | | |
| Software, algorithm | CellPhoneDB | https://www.cellphonedb.org/ | | |
| Software, algorithm | SCENIC | https://github.com/aertslab/SCENIC | | |
| Software, algorithm | ClusterProfiler | https://guangchuangyu.github.io/software/clusterProfiler/ | | |
| Software, algorithm | Slingshot | https://github.com/kstreet13/slingshot | | |
| Software, algorithm | scVelo | https://github.com/theislab/scvelo | | |
| Software, algorithm | FlowJo | https://www.flowjo.com | | |

## Animal husbandry

All mice were maintained according to the UCSF guidelines and the Laboratory Animal Resource Center standards. All experiments were reviewed and approved by the UCSF Institutional Animal Care and Use Committee. To generate all placentas C57Bl/6J males between 3 and 6 months of age were bred to C57Bl/6J females between 6 and 12 weeks of age (Jackson Laboratory, Bar Harbor, ME). Developmental staging was assessed by days post coitum (dpc) and confirmed by embryo morphology.

## Placental dissection

Each placenta sample was dissected in 1x PBS at 4°C. The yolk sac and allantois were removed from the basal chorionic plate using forceps. The outermost layers of the decidua were also removed, taking care to not disrupt the cells of the labyrinth or junctional zone of the fetal placenta. Sample passing these criteria were used in nuclei isolation below.

## Nuclei isolation

Whole dissected placental samples were placed in a 40 mm plastic dish with 1 mL of Nuclei EZ lysis buffer (Sigma-Aldrich, St.Louis, MO - NUC-101) and reduced to small pieces of tissues (<1 mm) using a razor blade. The solution containing the tissue was then transferred to a dounce homogenizer and 1 mL of Nuclei EZ lysis buffer added. The nuclei were lysed into solution by 10 strokes of the loose pestle (A) and 10 strokes of the tight pestle (B). Using a wide bore pipette the solution was transferred to a 15 mL falcon tube, 2 mL of Nuclei EZ lysis buffer added, and incubated on wet ice for 5 min with gentle mixing with a pipette. The solution was then passed through a 35 µM filter and centrifuged at 500 g for 5 min at 4°C. The resulting supernatant was removed, the pellet gently resuspended in 1.5 mL of Nuclei EZ lysis buffer and incubated on ice for 5 min, then centrifuged at 500 g for 5 min at 4°C. The supernatant was removed and 1 mL of Nuclei Wash and Resuspension Buffer (1x PBS, 1.0% BSA, 0.2 U/µL RNase Inhibitor) added without mixing or resuspension and incubated on ice for 5 min. After the incubation an additional 1 mL of Nuclei Wash and Resuspension Buffer was added and the pellet gently resuspended. The solution was then centrifuged at 500 g for 5 min at 4°C (in a swinging bucket rotor to prevent nuclei adhering to the sides of the tube), the supernatant discarded, and the nuclei resuspended in 1 mL of Nuclei Wash and Resuspension Buffer. This final nuclei solution was passed through a 35 µM filter and nuclei visually inspected on a hemocytometer to assess, morphology, damage, and aggregation. Finally, 1 µL of DAPI stock solution (10 µg/mL) was added to the nuclei solution in preparation for FACS.

## FAC-sorting of nuclei into 10x genomic reagents for transcriptome capture

All cell sorting was performed on a FACS Aria II (BD Biosciences, San Jose, CA) using a 70 µM nozzle. Nuclei sorting was performed according to steps outlined here - https://www.protocols.io/view/frankenstein-protocol-for-nuclei-isolation-from-f-3fkgjkw/abstract.

Nuclei were sorted in the nuclei wash and Resuspension Buffer (described above) plus DAPI. 17,500 nuclei yielding a strong DAPI signal and DNA content profile were sorted into one well of a 96-well plate containing 10x Genomics V3 GEM mastermix without RT Enzyme C (20.0 µL RT Reagent Mix, 3.1 µL Template Switch Oligo, 2.0 µL Reducing Agent B, 27.1 µL nuclease free H2O). The volume of 17,500 sorted nuclei was previously established to be 19.5 µL. Immediately post sorting 8.3 µL of RT Enzyme C (10x Genomics, Pleasanton, CA) was added to the sample, and transcriptome collection on the 10x Chromium V3 platform performed.

## Single-cell RNA sequencing and analysis

To capture the transcriptome of individual nuclei we used the Chromium Single Cell 3' Reagent V3 Kit from 10X Genomics. For all samples 17,500 nuclei were loaded into one well of a Chip B kit for GEM generation. Library preparation including, reverse transcription, barcoding, cDNA amplification, and purification was performed according to Chromium 10x V3 protocols. Each sample was sequenced on a NovaSeq 6000 S4 to a depth of approximately 20,000–30,000 reads per nucleus. The gene expression matrices for each dataset was generated using the CellRanger software (v3.0.2 - 10x Genomics). A custom premRNA GTF was generated to create an intron–exon reference

according to 10x Genomics recommendations and all reads aligned to mm10 using STAR. (https://support.10xgenomics.com/single-cell-gene-expression/software/pipelines/latest/advanced/references). The counts matrix was thresholded and analyzed in the package Seurat (v3.1.3). Nuclei with fewer than 500 or greater than 4000 unique genes, as well as all nuclei with greater than 0.25 percent mitochondrial counts, were excluded from all subsequent analyses. For each sample, counts were scaled and normalized using ScaleData and NormalizeData, respectively, with default settings and FindVariableFeatures used to identify the 2000 most variable genes as input for all future analyses. PCA was performed using RunPCA and significant PCs assessed using ElbowPlot and DimHeatmap. For individual analysis of each sample the number of PCs and the resolution parameter of FindClusters can be found in *Supplementary file 1*. Dimensionality reduction and visualization using UMAP was performed by RunUMAP. Differentially expressed genes were identified using FindAllMarkers. Integration of each timepoint into one dataset was performed using FindIntegrationAnchors and IntegrateData, both using 20 dimensions (after filtering each dataset for number of genes, mitochondrial counts, and normalizing as described above). Data scaling, PCA, selection of PCs, clustering and visualization proceeded as described above using 20 PCs and a resolution of 0.6. Each cluster was analyzed and clusters containing abnormal UMI and co-expression of multiple cell-type-specific genes were determined to be enriched for doublets and removed from downstream analysis. The number of nuclei excluded represented an estimated doublet rate of ~3.9%. To generate the trophoblast only dataset, data from each timepoint was subset for only trophoblast clusters using the function SubsetData based upon annotations from marker genes identified by FindAllMarkers. Integration of the trophoblast only dataset was performed using the same integration methods above (using 20 dimensions for FindIntegrationAnchors and IntegrateData, and 20 PCs and a resolution of 0.6 for FindClusters and RunUMAP). Differentially expressed genes for each integrated dataset were identified using FindAllMarkers.

## Slingshot

We used the Slingshot package to order cells for each SynTI, SynTII, and S-TGC lineage in pseudotime. Each lineage was subset and the 1000 most variable genes identified and used as input into the generalized linear model (GAM package). Expression is plotted for the top 200 most variable genes along each lineage and included in Supplementary data.

## scVelo

RNA velocity analysis was applied to the entire conglomerate dataset using Velocyto to generate spliced and unspliced reads for all cells. This dataset was then subset for the trophoblast dataset introduced in *Figure 2*. The scVelo stochastic model was run with default settings and subset by each timepoint. The magnitudes were calculated from the UMAP cell embeddings.

## Differential expression between timepoints

To account for the different number of nuclei recovered within a cluster at each timepoint, we downsampled to the least number of cells collected and performed differential expression using FindMarkers function and repeated this for 100 random permutations of nuclei. Only genes found to be significant (adj. p-value<0.05) in all 100 permutations were retained in the final list of differentially expressed genes. To retain the power in the larger clusters we did not downsample all clusters to the same size. Therefore the number of differentially expressed genes between timepoints cannot be compared between clusters.

## Gene ontology analysis

Gene Onotology analysis was performed with clusterProfiler enrichGO function. The simplify function within this package was used to consolidate hierarchically related terms using a cutoff of 0.5. Terms were considered significantly enriched with an adjusted P-value of less than 0.05.

## CellPhoneDB

Cluster annotations and counts data were used as input for CellPhoneDB with default settings (minimum of 10% of nuclei in a cluster expressing a gene, a p-value cutoff of 0.05, and 10 permutations). The databases of receptor–ligand interactions were generated for human proteins, not for mouse.

As such, we performed this analysis using orthogonal genes between species (a total of 724 genes in the CellPhoneDB receptor–ligand databases). However, this does not account for differences in receptor–ligand interactions that may be different between mouse and human. Raw data were analyzed and heatmaps generated in R using modified scripts from CellPhoneDB. For heatmaps, the mean values of each significant receptor–ligand pair are shown, and we do not consider non-significant interactions, these have been given the mean value of zero.

## Immunofluorescence

Placentas for cryosectioning were fixed in 4% PFA at 4°C for 8 hr, washed in 1x PBS, then submerged in 30% sucrose overnight at 4°C prior to embedding in OCT medium. Placentas were sectioned at 10 µM for all conditions. In brief, slides were washed in 1x PBST (1x PBS, 0.05% Tween-20), blocked for 1 hr (1x PBS 5% donkey serum + 0.3% TritonX), incubated in primary antibody diluted for 3 hr at room temperature (or overnight at 4°C), washed in 1x PBST, incubated in secondary antibody (Alexafluor 488, 594, and 680) for 1 hr at room temperature, incubated in DAPI for 10 min at room temp, washed in 1x PBST, and mounted and sealed for imaging. Any antigen retrieval was performed prior to the blocking step by heating the slides in a 1x citrate buffer with 0.05% Tween-20 at 95C for 30 min. All antibodies and the dilutions are listed in the key resources table. All immunofluorescence staining was performed in n = 3 biological replicates (three distinct placentas) and representative image is shown.

## Acknowledgements

We thank the members of the University of California, San Francisco National Center of Translational Research in Reproduction and Infertility for helpful comments during the design, execution, and publication of this project. A special thanks to Susan Fisher, Alex Pollen, and Peter Sudmant for insightful comments during data analysis and preparation on the manuscript. We would also like to thank all members of the Blelloch Lab for their comments and support, especially Brian DeVeale for help with data analysis and proofreading the manuscript. We would also like to acknowledge our funding source, the NIH Eunice Kennedy Shriver National Institute for Child Health and Human Development P50 HD055764.

## Additional information

### Funding

| Funder | Grant reference number | Author |
| --- | --- | --- |
| National Institute of Child Health and Human Development | P50 HD055764 | Robert Blelloch |

The funders had no role in study design, data collection and interpretation, or the decision to submit the work for publication.

### Author contributions

Bryan Marsh, Resources, Data curation, Formal analysis, Investigation, Visualization, Methodology, Writing - original draft; Robert Blelloch, Conceptualization, Supervision, Funding acquisition, Writing - review and editing

### Author ORCIDs

Bryan Marsh (iD) https://orcid.org/0000-0002-4979-5233
Robert Blelloch (iD) https://orcid.org/0000-0002-1975-0798

### Ethics

Animal experimentation: This study was performed in strict accordance with the recommendations in the Guide for the Care and Use of Laboratory Animals of the National Institutes of Health. All of

the animals were handled according to approved institutional animal care and use committee (IACUC) (Approval number: AN173513) of the University of California, San Francisco.

### Decision letter and Author response
Decision letter https://doi.org/10.7554/eLife.60266.sa1
Author response https://doi.org/10.7554/eLife.60266.sa2

## Additional files

### Supplementary files
• Supplementary file 1. Sample information and processing. Contains information of the number of nuclei captured at each timepoint and the processing information for each dataset (number of Principal Components and the resolution parameters used for cluster/integration)

• Supplementary file 2. Number of nuclei captured per cluster complete dataset. Breakdown of the number of nuclei collected at each timepoint for each cluster identified in the dataset used in *Figure 1*. Also, provided is the percent of the total nuclei assigned to each cluster captured at each timepoint. Finally, these data are normalized to the number of nuclei captured at each timepoint so that comparisons may be made with in a cluster across timepoints.

• Supplementary file 3. Number of nuclei captured per cluster trophoblast dataset. Breakdown of the number of nuclei collected at each timepoint for each cluster identified in the trophoblast dataset used in *Figures 2–6*. Also, provided is the percent of the total nuclei assigned to each cluster captured at each timepoint. Finally, these data are normalized to the number of nuclei captured at each timepoint so that comparisons may be made with in a cluster across timepoints.

• Transparent reporting form

### Data availability
Sequencing data have been deposited in GEO under accession code GSE152248. Processed data as R objects are available at figshare (https://figshare.com/projects/Single_nuclei_RNA-seq_of_mouse_placental_labyrinth_development/92354).

The following dataset was generated:

| Author(s) | Year | Dataset title | Dataset URL | Database and Identifier |
|---|---|---|---|---|
| Marsh BP, Blelloch RH | 2020 | Single nuclei RNA-seq of mouse placental labyrinth development | https://www.ncbi.nlm.nih.gov/geo/query/acc.cgi?acc=GSE152248 | NCBI Gene Expression Omnibus, GSE152248 |

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
