## [Decision Letter]

**Acceptance summary:**

This paper uses single nuclei sequencing of the developing mouse placenta. It gives particular insight into the differences in the syncytial layers of the labyrinth. It adds considerably to our knowledge of the sinusoidal trophoblast giant cells. The molecular detail provided is a substantial advance.

**Decision letter after peer review:**

Thank you for submitting your article "Single nuclei RNA-seq of the mouse placenta uncovers mechanisms of differentiation" for consideration by *eLife*. Your article has been reviewed by three peer reviewers, including Steve Charnock-Jones as the Reviewing Editor and Reviewer #1, and the evaluation has been overseen by Didier Stainier as the Senior Editor.

The reviewers have discussed the reviews with one another and the Reviewing Editor has drafted this decision to help you prepare a revised submission.

Summary:

This manuscript describes the use of single nuclei RNA sequencing of the mouse placenta. The authors performed snRNA-seq at four different stages of mouse embryonic development E9.5, E10.5, E12.5 and E14.5 and performed various computational analyses to identify distinct cells types of the developing placenta, especially, the progenitor and differentiated cell types of the placental labyrinth.

The use of separated nuclei allows analysis of the transcripts in the syncytial layers of the placenta to be analysed for the first time. This is a very significant advance and opens up a new dimension of analysis. The study provides information about possible new transcriptional regulatory mechanisms in placental cell types. The reviewers are therefore enthusiastic about this manuscript. As with other single nuclei or single cell papers this manuscript contains a very large amount of data. The quality and depth of these data will be extremely valuable to those working in the field. However, there are some areas of the manuscript that require revision and/or clarification.

Essential revisions:

1) In several places are there are claims that are too bold – they are not fully supported by the data and need to be toned down. For example: subsection “Developmental time course and trajectory inference reveal details of lineage dynamics and commitment” please change "luminogenesis" ->" lumenogenesis in early embryos". The inference of the role of Podxl is too strong. The lumenogenesis described by Shahbazi et al., is the formation of a lumen in the centre of a rosette of primitive endoderm whereas the lumen of the maternal blood spaces in the labyrinthine layer is formed from the invagination of the fused chorion and allantois by finger-like projections of extra-embryonic mesodermal cells. These processes are quite likely to be regulated differently. Similarly, the inference in subsection “Defining distinct roles of the trophoblast subtypes at the gas exchange interface” is too strong.

2) The authors seem to describe LaTP2 cells as "novel intermediate states". This is a bit of a stretch given the lack of information about the direct lineage relationship between the precursors and the differentiated cell types, or about how LaTP and LaTP2 might be related. In the absence of direct lineage analysis this text needs to be toned down to avoid overstating the findings (but please note the comment below about immunolocalization).

3) The discovery of LaTP2 cells is potentially very interesting and these cells seem to be largely defined by the absence of Met, Ror2, Lgr5. Can the authors rule out that this subpopulation is standing out simply because of technical dropout of these gene markers? The Uena et al., study showed that c-Met (and its ligand Hgf) sustain the LaTP population. Please comment on whether you think that the LaTP2 cells could be controlled through alternative pathways.

4) The analyses of hematopoietic cells using GATA4 and cKit is problematic. Multiple previous studies have shown that placental hematopoiesis starts at ~E10.5 as the hematopoietic stem cells emerge at that developmental stage and maximal number of HSCs are detected around E12.5. Thus, it is surprising that so many hematopoietic cells are detected at E9.5 and E10.5 and fewer at E12.5. Please reanalyze the data using additional hematopoietic markers.

5) The study does not provide information of Parietal giant cells and invasive trophoblast population and also contains incomplete information on other junctional zone cells. There are published reports that Junctional Zone cells, like glycogen cells undergo major changes in gene expression and function after E14.5 leading to the expansion of the junctional zone. (Coan et al., 2006). Also, Figure 3B clearly shows strong presence of junctional zone progenitors at E14.5, indicating incomplete differentiation. Thus, the study is not a complete picture of mouse placental development and authors should be more cautious about this aspect in the description of the title and within the body of the manuscript. Their description should be specific to their study, which is mostly focusing on the placental labyrinth zone involving SynT and S-TGC development and function.

6) Similarly, the authors need to explicitly comment on the cells that are poorly represented in their analysis. Specifically, the analyses of spongiotrophoblast vs. glycogen cells are also questionable. Ectoplacental cone derived cells expressing genes like Tpbp and Ascl2 are missing. These are well known genes with respect to the development of the spongiotrophoblast and progenitors of the invasive glycogen cells. The authors should include analyses with respect to these genes. In addition, some of their markers are also questionable. For example, Prl8a9 is highly expressed in Glycogen cells (Simmons et al., 2008). Also, authors should include data or comment about specific Glycogen cell markers like Connexin 31, IGF2R.

7) Subsection “Defining distinct roles of the trophoblast subtypes at the gas exchange interface” – While you can make the argument this cell layer functions as if it is a maternal endothelium I don't think it is correct to state "it forms a maternal endothelium". After all it is not composed of endothelial cells and does not have the receptors for the angiogenic growth factors made by SynTI and II as illustrated in Figure 5B. A description of the endothelial like properties of these cells would be warranted – indeed desirable. For example, do they have (nor not have) RNAs encoding anti- and adhesive surface markers (ICAM for example).

8) The section describing the likely functional differences in the trophoblast layers in the labyrinth requires additional detail and clarification (subsection “Defining distinct roles of the trophoblast subtypes at the gas exchange interface” and Figure 4). There seem to be three groups of "differentially expressed" genes – one for each of the trophoblast types (S-TGCs, SynTI and SynTII). It is not clear from the methods whether this is from all cells (albeit down-sampled) at all time points. I assume that as the FindMarkers function was used then the comparison is the group of interests (S-TGCs for example) vs all other groups. It would be helpful to have this detail clarified. Also, the authors should comment on why there appear not to be any genes that are lower in any group (so were only positive markers selected?) and what criteria were used to define a "differentially expressed" gene. In Supplementary file 14 there are apparently significantly regulated transcripts with a fold change of ~1.2 so are the differently expressed genes defined solely by the Padj? – but there is clearly some additional filtering (as shown in Figure 4C-E). Were all genes listed in Supplementary file 14 used for the ontogeny analysis? – the numbers of genes listed would suggest not (Supplementary file 14, 2104 genes; Supplementary file 15 686+483+842=1910). I presume not all the genes map to GO terms but the methods for the GO analysis are not described so it is not possible to know this.

9) The GO analysis yields a very large number of significant terms with the top 5 clusters for each cell type presented in Figure 4C-E. The rationale for choosing the "top 5" is not clear – especially as clusters (ie GO terms) are hierarchically related. This is exemplified in 4D where cluster D (GO:0072659) is actually a child of term B (GO:1990778). Finally, the large number of terms reported is hard to understand as they are from different levels in the GO term hierarchy and report the same pathway (eg hippo signalling and regulation of hippo signalling).

10) There is very little said about the possible function and development of the S-TGCs. These cells (as their name implies) have previously been described as being similar to giant cells. The UMAP shown in Figure 2A suggests they arise from a population of cells related to the junctional zone precursors. Indeed, this would fit with the prolactin expression profiles (see Simmons et al., 2008). Notably, the prolactins, which are present in the S-TGCs are not in found in the labyrinthine syncytial cells but in EPC derived cells lineage. This requires some discussion.

11) The analysis of likely ligand – receptor interactions, and the transcription factor regulons, raise interesting new leads, but very few are validated.

12) As authors are proposing presence of a new progenitor population, it is important to provide more information about the origin, timing of emergence as well as spatial distribution of the two LaTP progenitor lineages. These aspects need more analyses, especially at an earlier developmental time point (E7.5-E9). As the authors have identified probable unique marker genes, they should include additional studies with placental sections during early time points (E7.5-E9) to better characterize the two LaTP populations.

To address these concerns (11-12), we request some additional immunolocalization studies be added. Of particular interest are transcription factors and markers that differentiate between the LaTP and LaTP2 lineages. We appreciate this this work will be constrained by the availability of suitable antibodies.

13) The two syncytial layers for a continuous barrier between the maternal and fetal compartments. Nutrient, gas and waste products have to cross or be transported across this barrier – ie they need to cross both layers. The section describing this (subsection “Defining distinct roles of the trophoblast subtypes at the gas exchange interface”) does not address this. Indeed, the analytical approach seems flawed – by focusing on the genes that are "enriched" in one layer, overlooks that fact that a transport network may exist across both layers. Of course, it is quite possible that one mechanism may be used to transport something across one layer and a different mechanism operate for the other layer, nonetheless the nutrients needs to traverse both layers. Describing the transporter RNAs that are present in both SynT layers is essential. This is an opportunity that has been missed as this study has the resolution to evaluate this for the first time.

---

## [Author Response]

Summary:This manuscript describes the use of single nuclei RNA sequencing of the mouse placenta. The authors performed snRNA-seq at four different stages of mouse embryonic development E9.5, E10.5, E12.5 and E14.5 and performed various computational analyses to identify distinct cells types of the developing placenta, especially, the progenitor and differentiated cell types of the placental labyrinth.The use of separated nuclei allows analysis of the transcripts in the syncytial layers of the placenta to be analysed for the first time. This is a very significant advance and opens up a new dimension of analysis. The study provides information about possible new transcriptional regulatory mechanisms in placental cell types. The reviewers are therefore enthusiastic about this manuscript. As with other single nuclei or single cell papers this manuscript contains a very large amount of data. The quality and depth of these data will be extremely valuable to those working in the field. However, there are some areas of the manuscript that require revision and/or clarification.

We thank the reviewers for their interest and insightful suggestions to improve this manuscript. We have added additional analyses and validation of findings with immunofluorescence staining in the main and supplementary figures. Additionally, we have incorporated the reviewers’ feedback and modified the text for clarity and to better reflect the data presented. The specific changes are detailed below.

Essential revisions:1) In several places are there are claims that are too bold – they are not fully supported by the data and need to be toned down. For example: subsection “Developmental time course and trajectory inference reveal details of lineage dynamics and commitment” please change "luminogenesis" ->" lumenogenesis in early embryos". The inference of the role of Podxl is too strong. The lumenogenesis described by Shahbazi et al. is the formation of a lumen in the centre of a rosette of primitive endoderm whereas the lumen of the maternal blood spaces in the labyrinthine layer is formed from the invagination of the fused chorion and allantois by finger-like projections of extra-embryonic mesodermal cells. These processes are quite likely to be regulated differently. Similarly, the inference in subsection “Defining distinct roles of the trophoblast subtypes at the gas exchange interface” is too strong.

The sentences concerning the formation of lumen structures in the early embryo and the maternal blood sinuses in the placenta have been amended to reflect that these are different processes. Podxl likely does not function the same way in both contexts. We have changed the text to clarify that the expression and charge of Podxl may function to maintain open blood sinuses, not generate sinuses de novo.

2) The authors seem to describe LaTP2 cells as "novel intermediate states". This is a bit of a stretch given the lack of information about the direct lineage relationship between the precursors and the differentiated cell types, or about how LaTP and LaTP2 might be related. In the absence of direct lineage analysis this text needs to be toned down to avoid overstating the findings (but please note the comment below about immunolocalization).

“Novel intermediate States” is meant to reflect the SynTI Precursor, SynTII Precursor, and S-TGC Precursor populations, which have distinct gene expression from both progenitors and their respective differentiated states. The text has been modified to reflect this. In concert with Essential revision point 12, we have provided additional evidence for gene expression differences between LaTP and LaTP2. Between the LaTP populations, we identify Met as largely restricted to LaTP and Egfr largely restricted to LaTP2 (Figure 2—figure supplement 1C). We colocalize each protein with EPCAM, an established marker of LaTP (previously described in Ueno et al., 2013) which through sparse clonal labeling has been shown to give rise to SynTI, SynTII, and S-TGC. We identify at E9.5 that MET and EPCAM have matching domains of expression, while EGFR and EPCAM are largely non-overlapping, although rare double positive cells do exist (Figure 2—figure supplement 1D). Similar to mRNA, EGFR and MET protein appear to be mutually exclusive. The MET/EPCAM positive cells appear more basally localized than EGFR positive cells suggestive that the former gives rise to the later. The new analysis and immunofluorescence data have been added to the supplementary figures and the text has been changed to reflect these findings.

3) The discovery of LaTP2 cells is potentially very interesting and these cells seem to be largely defined by the absence of Met, Ror2, Lgr5. Can the authors rule out that this subpopulation is standing out simply because of technical dropout of these gene markers? The Uena et al., study showed that c-Met (and its ligand Hgf) sustain the LaTP population. Please comment on whether you think that the LaTP2 cells could be controlled through alternative pathways.

The data in Figure 1—figure supplement 1B and C and Figure 2- 1A demonstrate that LaTP and LaTP2 do not differ drastically in the number of unique genes identified, the number of total transcripts identified, or the percent of mitochondrial reads. Thus, we believe the differences in these populations is not due to technical dropout or low information cell driving the clustering. We have identified Egfr as a strong marker of the LaTP2 population (Figure 2—figure supplement 1C and D). We also confirm strong MET protein expression in the LaTP population. Because LaTP2 do not express Met or Wnt receptors, Ror2 and Lgr5, it is unlikely this population is sustained by HGF-MET or Wnt signaling. However, given the expression of Egfr (both transcript and protein) in LaTP2 and the expression of Egf ligands, Tgfa and Hbegf, by SynTI we hypothesize that Egf signaling is important for control of LaTP2. The text and figures (Figure 2—figure supplement 1B-E) have been updated to include these new data.

4) The analyses of hematopoietic cells using GATA4 and cKit is problematic. Multiple previous studies have shown that placental hematopoiesis starts at ~E10.5 as the hematopoietic stem cells emerge at that developmental stage and maximal number of HSCs are detected around E12.5. Thus, it is surprising that so many hematopoietic cells are detected at E9.5 and E10.5 and fewer at E12.5. Please reanalyze the data using additional hematopoietic markers.

We included clusters 15 and 19 in the Fetal Mesenchyme group due to their mesenchymal features, but did note these cell expressed some HSC markers but lacked other important markers. We analyzed the gene expression in these clusters according to markers found in McKinney-Freeman et al., 2013. No cluster identified in our data matches with HSCs, although other hematopoietic lineages express some of these markers. Recent, single cell sequencing data in the mouse identified HSCs to be only 0.05% of total cells. By these numbers we would only expect 13 HSC nuclei in our data. Likely, the lack of HSCs in our data is due to the rarity of this population.

We have edited the text to clarify the annotation of these cells in mesenchyme (not trophoblast and not HSC). Additionally, we performed immunofluorescence staining for another shared marker of clusters 15 and 19, PDPN. This shows expression in interstitial cells with long processes which seem to make contact with fetal endothelial cells. These data are now included in Figure 1—figure supplement 1E and mentioned in text.

5) The study does not provide information of Parietal giant cells and invasive trophoblast population and also contains incomplete information on other junctional zone cells. There are published reports that Junctional Zone cells, like glycogen cells undergo major changes in gene expression and function after E14.5 leading to the expansion of the junctional zone. (Coan et al., 2006). Also, Figure 3B clearly shows strong presence of junctional zone progenitors at E14.5, indicating incomplete differentiation. Thus, the study is not a complete picture of mouse placental development and authors should be more cautious about this aspect in the description of the title and within the body of the manuscript. Their description should be specific to their study, which is mostly focusing on the placental labyrinth zone involving SynT and S-TGC development and function.

The title and descriptions of the study have been changed to reflect the focus on the development of the placental labyrinth and aspects not included in this study (maturation of the junctional zone after E14.5) are explicitly listed.

6) Similarly, the authors need to explicitly comment on the cells that are poorly represented in their analysis. Specifically, the analyses of spongiotrophoblast vs. glycogen cells are also questionable. Ectoplacental cone derived cells expressing genes like Tpbp and Ascl2 are missing. These are well known genes with respect to the development of the spongiotrophoblast and progenitors of the invasive glycogen cells. The authors should include analyses with respect to these genes. In addition, some of their markers are also questionable. For example, Prl8a9 is highly expressed in Glycogen cells (Simmons et al., 2008). Also, authors should include data or comment about specific Glycogen cell markers like Connexin 31, IGF2R.

We used Pcdh12 as a marker of glycogen cell identity (Simmons et al., 2008). In our data Pcdh12 is expressed in increasing levels from JZP1 to JZP2 to differentiated glycogen cells. We thank the reviewers for suggesting Simmons et al., BMC Genomics 2008 as a resource for determining Glycogen Cell vs. Spongiotrophoblast specific expression of prolactin genes. We analyzed the expression of the prolactin genes highlighted in this paper in our dataset. This data is now included in Figure 2-figure supplement 3A and B, alongside the expression of Pcdh12 and the other markers suggested by the reviewers (Tpbpa, Ascl2, Connexin31/Gjb3, and Igf2r). Our data is largely in agreement with the in situ hybridization data from Simmons et al. 2008, and this is summarized in Figure 2-figeru supplement3. While it is clear that there are several common markers of the SpT and Glycogen trophoblast and that prolactin expression in the placenta is quite complex, we feel the annotation we have provided are the best possible considering the current understanding of the field.

7) Subsection “Defining distinct roles of the trophoblast subtypes at the gas exchange interface”- While you can make the argument this cell layer functions as if it is a maternal endothelium I don't think it is correct to state "it forms a maternal endothelium". After all it is not composed of endothelial cells and does not have the receptors for the angiogenic growth factors made by SynTI and II as illustrated in Figure 5B. A description of the endothelial like properties of these cells would be warranted – indeed desirable. For example, do they have (nor not have) RNAs encoding anti- and adhesive surface markers (ICAM for example).

We analyzed these cells for specific endothelial markers and do not see them highly expressed. The language has been changed to reflect that these cells form a lining but lack canonical endothelial properties.

8) The section describing the likely functional differences in the trophoblast layers in the labyrinth requires additional detail and clarification (subsection “Defining distinct roles of the trophoblast subtypes at the gas exchange interface” and Figure 4). There seem to be 3 groups of "differentially expressed" genes – one for each of the trophoblast types (S-TGCs, SynTI and SynTII). It is not clear from the methods whether this is from all cells (albeit down-sampled) at all time points. I assume that as the FindMarkers function was used then the comparison is the group of interests (S-TGCs for example) vs all other groups. It would be helpful to have this detail clarified. Also, the authors should comment on why there appear not to be any genes that are lower in any group (so were only positive markers selected?) and what criteria were used to define a "differentially expressed" gene. In Supplementary file 14 there are apparently significantly regulated transcripts with a fold change of ~1.2 so are the differently expressed genes defined solely by the Padj? – but there is clearly some additional filtering (as shown in Figure 4C-E). Were all genes listed in Supplementary file 14 used for the ontogeny analysis? – the numbers of genes listed would suggest not (Supplementary file 14, 2104 genes; Supplementary file 15 686+483+842=1910). I presume not all the genes map to GO terms but the methods for the GO analysis are not described so it is not possible to know this.

For these analyses the cells in the SynTI, SynTII, and S-TGC clusters were subset from the trophoblast dataset containing all timepoints. FindAllMarkers was used to identify positive differentially expressed genes among these three populations (one population against the other two, for each population). Differentially expressed genes are determined using a Wilcoxon rank sum test considering genes expressed in at least 25% of cells in a cluster and those with log fold change of 0.25 or greater.

All markers were included in the GO analysis using the package clusterProfiler. The list of genes was first converted to Entrez IDs. In each cluster approximately 1% of genes did not match to Entrez IDs (SynTI – 7/715; SynTII – 8/507; S-TGC – 12/882). Transcripts that did not match represent non-coding RNAs and predicted genes. Those genes with Entrez IDs were then subject to GO analysis and for each cluster with 95-96% of genes mapping to at least one GO categories (SynTI – 686/708; SynTII – 482/499; S-TGC – 842/870). Therefore, the analysis considers 2010 out of a possible 2104 marker genes (95.5%). The methods have been updated with a thorough description of the GO analysis.

9) The GO analysis yields a very large number of significant terms with the top 5 clusters for each cell type presented in Figure 4C-E. The rationale for choosing the "top 5" is not clear – especially as clusters (ie GO terms) are hierarchically related. This is exemplified in 4D where cluster D (GO:0072659) is actually a child of term B (GO:1990778). Finally, the large number of terms reported is hard to understand as they are from different levels in the GO term hierarchy and report the same pathway (eg hippo signalling and regulation of hippo signalling).

We thank the reviewers for pointing out the redundancy. We have repeated this analysis, this time using a cutoff to consolidate hierarchically related terms with greater similarity than 0.5. We still show the top 5 GO terms ranked by adjusted p-value for clarity of presentation. However, the top 10 GO terms are included in Figure 4—figure supplement 1A and all significantly enriched GO terms are included in Supplementary Table 15. If reviewers are aware of alternative means/packages for consolidating related terms, we would be happy to apply them. For now, we present our latest analysis in Figure 4C-E and Figure 4—figure supplement 1A have updated the text accordingly.

10) There is very little said about the possible function and development of the S-TGCs. These cells (as their name implies) have previously been described as being similar to giant cells. The UMAP shown in Figure 2A suggests they arise from a population of cells related to the junctional zone precursors. Indeed, this would fit with the prolactin expression profiles (see Simmons et al., 2008). Notably, the prolactins, which are present in the S-TGCs are not in found in the labyrinthine syncytial cells but in EPC derived cells lineage. This requires some discussion.

The reviewer’s raise an interesting point concerning the origin of S-TGCs. Simmons et al., 2007 and Simmons et al., 2011 provide evidence for these cells arising from either ectoplacental cone or chorion trophoblast. Ueno et al. 2013 demonstrate S-TGC can arise from Epcam^hi^ LaTP (a chorionic population). Our data place the S-TGC as sprouting from between the LaTP and JZP populations. Unfortunately, this does not conclusively provide an answer to the origin of S-TGC, but supports idea that this population may arise from multiple locations. Additional discussion of the genesis has been included in the text describing Figure 2, and additional discussion of function has been added to the discussion of Figure 4.

11) The analysis of likely ligand – receptor interactions, and the transcription factor regulons, raise interesting new leads, but very few are validated.

In addition to the validation of VEGFA protein expression in SynTII cells (Figure 5B and Figure 5—figure supplement 2A), we also validated the expression of HBEGF in SynTI through colocalization with SynTI marker STRA6 (Figure 5—figure supplement 2B). We further confirm Egf signaling between SynTI and SynTII and colocalized the ligand HBEGF and the receptor EGFR. These experiments reveal only HBEGF protein in SynTI but co-expression of HBEGF and EGFR in SynTII, suggesting functional signaling in this layer (Figure 5D and Figure 5—figure supplement 2C).

To validate the expression of a SCENIC predicted TF regulon, we selected GATA1. Staining for GATA1 revealed expression in trophoblast to be restricted to SynTII, although GATA1 is strongly expressed by several hematopoietic lineages in the fetal vessels (Figure 6F and Figure 6—figure supplement 1D). Colocalization of GATA1 with SynTII marker IGF1R demonstrates GATA1 positive nuclei abutting the basal membrane of SynTII marked by IGF1R.

12) As authors are proposing presence of a new progenitor population, it is important to provide more information about the origin, timing of emergence as well as spatial distribution of the two LaTP progenitor lineages. These aspects need more analyses, especially at an earlier developmental time point (E7.5-E9). As the authors have identified probable unique marker genes, they should include additional studies with placental sections during early time points (E7.5-E9) to better characterize the two LaTP populations.To address these concerns (11-12), we request some additional immunolocalization studies be added. Of particular interest are transcription factors and markers that differentiate between the LaTP and LaTP2 lineages. We appreciate this this work will be constrained by the availability of suitable antibodies.

The reviewers raise important points concerning the emergence and location of the two LaTP populations. To address this question, we have now done additional immunostaining as described above. To recap: first, we identify Met as a specific marker of LaTP, and Egfr as a highly enriched marker in LaTP2 (Figure 2—figure supplement 1B and C). We then colocalized each of these proteins with EPCAM, the canonical LaTP marker identified by Ueno et al., 2013 (Figure 2—figure supplement 1D). At E9.5 EPCAM and MET have overlapping domains of expression. This is in agreement with the transcript data and suggests that the LaTP cluster is the LaTP population as defined by Ueno et al., 2013. In contrast, only the rare EGFR+ cell showed coexpression with EPCAM protein (Figure 2—figure supplement 1D). Interestingly the EGFR+ cells were juxtaposed to the MET+ and positioned as such to suggest that the later may be giving rise to the former.

In addition to the staining at E9.5, we also added staining at E8.5 (Figure 2—figure supplement 1E). This timepoint is coincident with chorioallantoic fusion. We found many EPCAM/MET positive cells in the chorion, but few to no EGFR expressing cells. These data would be consistent with interpretation that EPCAM/MET precede and may even give rise to the EGFR+ LaTP2 cells. Along with figures noted above, we have added text to discuss these findings.

13) The two syncytial layers for a continuous barrier between the maternal and fetal compartments. Nutrient, gas and waste products have to cross or be transported across this barrier – ie they need to cross both layers. The section describing this (subsection “Defining distinct roles of the trophoblast subtypes at the gas exchange interface”) does not address this. Indeed, the analytical approach seems flawed – by focusing on the genes that are "enriched" in one layer, overlooks that fact that a transport network may exist across both layers. Of course, it is quite possible that one mechanism may be used to transport something across one layer and a different mechanism operate for the other layer, nonetheless the nutrients needs to traverse both layers. Describing the transporter RNAs that are present in both SynT layers is essential. This is an opportunity that has been missed as this study has the resolution to evaluate this for the first time.

We apologize for the lack of clarity concerning this analysis. The analysis of transport genes includes all genes found to differentially expressed in any of the interface populations (SynTI, SynTII, and/or S-TGC) compared to all other trophoblast populations. However, since this is distinct from the differentiation expression shown in Figure 4B and used in the GO analysis in Figure 4C-E. The text has been changed to explicitly state which genes were analyzed. This information is also included in the figure legend.